# Coherent-state path integrals in quantum thermodynamics

**Luca Salasnich[1,2,3,4]⋆ and Cesare Vianello[1,2]†**

**1** Dipartimento di Fisica e Astronomia "Galileo Galilei", Università di Padova,
Via Marzolo 8, I-35131 Padova, Italy
**2** INFN Sezione di Padova, Via Marzolo 8, I-35131 Padova, Italy
**3** Padua QTech Center, Università di Padova, Via Gradenigo 6/A, I-35131 Padova, Italy
**4** CNR-INO, Via Carrara 1, I-50019 Sesto Fiorentino, Italy

⋆ luca.salasnich@unipd.it , † cesare.vianello@phd.unipd.it

## Abstract

In these notes, we elucidate some subtle aspects of coherent-state path integrals, focusing on their application to the equilibrium thermodynamics of quantum many-particle systems. These subtleties emerge when evaluating path integrals in the continuum, either in imaginary time or in Matsubara-frequency space. Our central message is that, when handled with due care, the path integral yields results identical to those obtained from the canonical Hamiltonian approach. We illustrate this through a pedagogical treatment of several paradigmatic systems: the bosonic and fermionic harmonic oscillators, the single-site Bose-Hubbard and Hubbard models, the weakly-interacting Bose gas with finite-range interactions, and the BCS superconductor with finite-range interactions.

# 1 Introduction

The central problem of equilibrium (quantum) thermodynamics is the determination of the thermodynamic potential, namely, the Helmholtz free energy in the canonical ensemble or the grand potential in the grand canonical ensemble. In both cases, the thermodynamic potential is proportional to the logarithm of the partition function, which encodes the full equilibrium properties of the system. Hence, the accurate evaluation of the partition function lies at the heart of any theoretical description of quantum many-particle systems at finite temperature. The direct route to the partition function consists in evaluating the trace of the density operator, which involves the exponential of the Hamiltonian. Because the Hamiltonians of interacting many-particle systems contains non-commuting operators, this procedure is typically limited to perturbative treatments, giving rise to diagrammatic methods [1]. An alternative route to evaluating the partition function, and to constructing the associated perturbative and diagrammatic expansions, is provided by the coherent-state path integral. By expressing the partition function as a functional integral over complex- or Grassmann-valued fields, this formalism forges a natural bridge between the canonical (Hamiltonian) description and the field-theoretic methods that are central to modern theoretical physics.

While many standard textbooks are devoted to the path-integral formalism [2–8], subtle technical points are often treated only briefly in favor of practical applications, understandably so, yet this can leave students (and not only them [9–12]) with lingering confusion. These subtleties become evident when taking the continuum limit, whether in imaginary time or in the Matsubara-frequency representation, and concern, in particular, the correct treatment of variable transformations, functional determinants, and the regularization of Matsubara-frequency summations. Neglecting such details may lead to discrepancies between results obtained from the path-integral formalism and those derived from the canonical Hamiltonian approach, even for the simplest systems.

The purpose of these notes is to discuss in detail these subtleties and to show that, when handled correctly, the coherent-state path integral yields results fully consistent with the canonical formalism. We present a unified account that emphasizes both the logical coherence of the formalism and the common sources of error. To this end, we develop a series of examples of increasing complexity, beginning with the bosonic and fermionic harmonic oscillators and their immediate extensions to the single-site Bose-Hubbard and Hubbard models, and proceeding to the weakly interacting Bose gas and the Bardeen-Cooper-Schrieffer (BCS) superconductor, the two paradigmatic models for ultracold quantum systems in the continuum. For these systems, we also generalize both the Hamiltonian and the path integral formalisms

to include finite-range interactions, an aspect rarely treated at a pedagogical level. Each example serves a dual role: it illustrates the technical aspects of constructing and evaluating coherent-state path integrals in the continuum, and it clarifies the conceptual equivalence between the path-integral and operator-based descriptions of equilibrium thermodynamics. Due to this focus, the discussion emphasizes the technical aspects of the computations rather than the physical interpretation or consequences of the quantities obtained, which are however extensively reviewed in the standard textbooks. In doing so, we aim to provide a pedagogical yet rigorous resource for students seeking a transparent and internally consistent treatment of coherent-state path integrals in quantum many-particle physics.

## 2 Coherent-state path integrals

We begin by reviewing the standard construction of the coherent-state path integral, following Refs. [2–8].

### 2.1 Bosonic path integrals

Consider a Hamiltonian $\hat{H}$ involving a single pair of bosonic creation and annihilation operators $\hat{a}^\dagger$, $\hat{a}$ satisfying the canonical commutation relations

$$[\hat{a}, \hat{a}^\dagger] = 1, \qquad [\hat{a}, \hat{a}] = [\hat{a}^\dagger, \hat{a}^\dagger] = 0, \tag{2.1}$$

where $[\hat{A}, \hat{B}] \equiv \hat{A}\hat{B} - \hat{B}\hat{A}$. The Hilbert space is generated by the algebra of the creation and annihilation operators acting on the vacuum state $|0\rangle$ defined by $\hat{a}|0\rangle = 0$, and as such it has an overcomplete basis constituted by the *coherent states* $|a\rangle$, defined as the eigenstates of the annihilation operator,

$$\hat{a}|a\rangle = a|a\rangle. \tag{2.2}$$

These satisfy

$$|a\rangle = e^{a\hat{a}^\dagger}|0\rangle, \tag{2.3a}$$

$$\langle a|a'\rangle = e^{a^*a'}, \tag{2.3b}$$

$$\int \frac{da^* da}{2\pi i} \, e^{-a^*a} |a\rangle\langle a| = \hat{1}, \tag{2.3c}$$

where $a^*$, $a$ are *complex conjugate numbers* and $da^* da/2\pi i = d(\mathrm{Re}\,a)d(\mathrm{Im}\,a)/\pi$. As a consequence of the completeness relation (2.3c), the trace of an any operator $\hat{Q} = Q(\hat{a}, \hat{a}^\dagger)$ can be written as

$$\mathrm{Tr}\,\hat{Q} = \int \frac{da^* da}{2\pi i} \, e^{-a^*a} \langle a|\hat{Q}|a\rangle. \tag{2.4}$$

The canonical partition function of the system may thus be written as

$$\mathcal{Z} = \mathrm{Tr}\left(e^{-\beta\hat{H}}\right) = \int \frac{da^* da}{2\pi i} \, e^{-a^*a} \langle a|e^{-\beta\hat{H}}|a\rangle, \tag{2.5}$$

where $\beta$ is the reciprocal of the thermodynamic temperature. Inserting $M - 1$ resolutions of the identity at equally-spaced imaginary-time intervals of length $\delta\tau \equiv \beta\hbar/M$, we get

$$\mathcal{Z} = \int \left(\prod_{j=1}^{M} \frac{da_j^* da_j}{2\pi i}\right) e^{-\sum_{j=1}^{M} a_j^* a_j} \prod_{j=1}^{M} \langle a_j|e^{-\frac{\delta\tau}{\hbar}\hat{H}}|a_{j-1}\rangle, \tag{2.6}$$

with the identification $a = a_M = a_0$.

Let us take a closer look at an individual term in the product on the right-hand side of Eq. (2.6). For definiteness, consider the simple case of a Gaussian Hamiltonian $\hat{H} = \varepsilon \hat{a}^\dagger \hat{a}$, with $\varepsilon > 0$. This can be viewed either as a noninteracting many-particle system with a single available energy level $\varepsilon$, or as a 1D harmonic oscillator (without the zero-point energy), where the many particles correspond to its quasiparticle excitations. Using the definition of the exponential, we can write $\langle a_j | e^{-\frac{\delta\tau}{\hbar}\hat{H}} | a_{j-1} \rangle = \langle a_j | e^{-\tilde{\varepsilon}\hat{a}^\dagger \hat{a}} | a_{j-1} \rangle = \sum_{p=0}^\infty \frac{(-\tilde{\varepsilon})^p}{p!} \langle a_j | (\hat{a}^\dagger \hat{a})^p | a_{j-1} \rangle$, where $\tilde{\varepsilon} \equiv \beta\varepsilon/M$. By using the commutation relations (2.1), the $p$-th power of $\hat{a}^\dagger \hat{a}$ can be written in normal-ordered form, i.e. with all creation operators to the left of all annihilation operators, as $(\hat{a}^\dagger \hat{a})^p = \sum_{k=0}^p \left\{ {p \atop k} \right\} (\hat{a}^\dagger)^k \hat{a}^k$, where $\left\{ {p \atop k} \right\}$ are Stirling numbers of the second kind, having the combinatorial interpretation of the number of partitions of a set of $p$ objects into $k$ non-empty subsets [13]. Therefore

$$\langle a_j | e^{-\tilde{\varepsilon}\hat{a}^\dagger \hat{a}} | a_{j-1} \rangle = \langle a_j | a_{j-1} \rangle \sum_{p=0}^\infty \frac{(-\tilde{\varepsilon})^p}{p!} \sum_{k=0}^p \left\{ {p \atop k} \right\} (a_j^* a_{j-1})^k. \tag{2.7}$$

Since $\tilde{\varepsilon} > 0$, this double series is absolutely convergent, and we can swap the two summations to obtain[1]

$$\begin{aligned}
\langle a_j | e^{-\tilde{\varepsilon}\hat{a}^\dagger \hat{a}} | a_{j-1} \rangle &= \langle a_j | a_{j-1} \rangle \sum_{k=0}^\infty (a_j^* a_{j-1})^k \sum_{p=k}^\infty \left\{ {p \atop k} \right\} \frac{(-\tilde{\varepsilon})^p}{p!} \\
&= \langle a_j | a_{j-1} \rangle \sum_{k=0}^\infty (a_j^* a_{j-1})^k \frac{(e^{-\tilde{\varepsilon}} - 1)^k}{k!} \\
&= \langle a_j | a_{j-1} \rangle \exp\left[ a_j^* a_{j-1} (e^{-\tilde{\varepsilon}} - 1) \right],
\end{aligned} \tag{2.8}$$

where in the second line we used the known closed form for the inner $p$-sum (the exponential generating function of the Stirling numbers of the second kind). If $\tilde{\varepsilon}$ is sufficiently small, i.e. $M$ is sufficiently large, we can expand this exact result up to first order in $\tilde{\varepsilon}$, obtaining

$$\langle a_j | e^{-\tilde{\varepsilon}\hat{a}^\dagger \hat{a}} | a_{j-1} \rangle = \langle a_j | a_{j-1} \rangle e^{-\tilde{\varepsilon} a_j^* a_{j-1}} + O(\tilde{\varepsilon}^2). \tag{2.9}$$

This is the building block of the coherent-state path integral. The same argument applies to a generic Hamiltonian $\hat{H} = H(\hat{a}, \hat{a}^\dagger)$ after it has been put in *normal-ordered* form using the commutation relations. Using Eq. (2.3b), we thus get

$$\langle a_j | e^{-\frac{\delta\tau}{\hbar}\hat{H}} | a_{j-1} \rangle = e^{a_j^* a_{j-1} - \frac{\delta\tau}{\hbar} H(a_j^*, a_{j-1})} + O(\delta\tau^2). \tag{2.10}$$

Substituting this into Eq. (2.6) then yields $\mathcal{Z} = \mathcal{Z}_M + O(\delta\tau^2)$, where

$$\begin{aligned}
\mathcal{Z}_M &= \int \left( \prod_{j=1}^M \frac{da_j^* da_j}{2\pi i} \right) e^{-\sum_{j=1}^M a_j^* a_j} e^{\sum_{j=1}^M \left[ a_j^* a_{j-1} - \frac{\delta\tau}{\hbar} H(a_j^*, a_{j-1}) \right]} \\
&= \int \left( \prod_{j=1}^M \frac{da_j^* da_j}{2\pi i} \right) e^{-\frac{\delta\tau}{\hbar} \sum_{j=1}^M \left[ \hbar a_j^* \frac{a_j - a_{j-1}}{\delta\tau} + H(a_j^*, a_{j-1}) \right]}
\end{aligned} \tag{2.11}$$

is the discretized coherent-state path integral representation of the partition function[2]. Taking the limit $M \to \infty$, $\delta\tau \to 0$, $\delta\tau M = \beta\hbar$, this converges to the exact the partition function,

$$\lim_{M \to \infty} \mathcal{Z}_M = \mathcal{Z}. \tag{2.12}$$

---

[1] Considering the partial sum up to $p = L$, it is easy to verify that one can reorder the finite sums as $\sum_{p=0}^L \frac{(-\tilde{\varepsilon})^p}{p!} \sum_{k=0}^p \left\{ {p \atop k} \right\} (a_j^* a_{j-1})^k = \sum_{k=0}^L (a_j^* a_{j-1})^k \sum_{p=k}^L \left\{ {p \atop k} \right\} \frac{(-\tilde{\varepsilon})^p}{p!}$, which is just a reindexing identity. The fact that the equivalence remains valid in the limit $P \to \infty$ is guaranteed by the absolute convergence of the double series.

[2] We notice that the term $a_j^*(a_j - a_{j-1})$ at the exponent of Eq. (2.11) can be written equivalently in the symmetric form $[a_j^*(a_j - a_{j-1}) - (a_j^* - a_{j-1}^*)a_{j-1}]/2$.

The exponent of Eq. (2.11) is the discrete-time version of the classical Euclidean action

$$S[a^*, a] = \int_0^{\beta\hbar} d\tau \left\{ a^*(\tau)\hbar\partial_\tau a(\tau) + H(a^*, a) \right\}, \tag{2.13}$$

where

$$H(a^*, a) \equiv \frac{\langle a|\hat{H}|a\rangle}{\langle a|a\rangle} \tag{2.14}$$

is the expectation value of the normal-ordered Hamiltonian on the bosonic coherent state, and $a^*(\tau)$, $a(\tau)$ are complex-valued functions, periodic of period $\beta\hbar$. Therefore we will also write $\mathcal{Z}$ as the continuous imaginary-time path integral

$$\mathcal{Z} = \int_{a(\beta\hbar)=a(0)} \mathcal{D}a^*\mathcal{D}a \, e^{-S[a^*, a]/\hbar}, \tag{2.15}$$

where

$$\mathcal{D}a^*\mathcal{D}a \equiv \lim_{M\to\infty} \prod_{j=1}^{M} \frac{da_j^* da_j}{2\pi i}. \tag{2.16}$$

At this point it is worth mentioning that different operator orderings (e.g. Weyl ordering, anti-normal ordering, etc.) are also possible, which correspond to different discretizations of the coherent-state path integral [14]. Each discretized path integral represents the same quantum Hamiltonian, and all orderings are physically equivalent in the sense that, if each discretized partition function is computed with its own time-slicing rule and the continuum limit $M \to \infty$ is taken at the end, they all reproduce the same partition function. Different discretizations come with different prescriptions in the continuum limit, namely a specific form for the Hamiltonian symbol $H(a^*, a)$, a specific prescription for equal-time operator products, and the associated expressions for functional determinants. Among the possible choices, the normal-ordered action is distinguished in that it admits a simple interpretation of the Hamiltonian symbol $H(a^*, a)$ in the continuum as the expectation value of the quantum Hamiltonian on the coherent state.

The construction presented above readily extends to the many-particle case. A many-particle bosonic Hamiltonian involves a complete set of annihilation operators $\{\hat{a}_\alpha\}$ and the corresponding creation operators $\{\hat{a}_\alpha^\dagger\}$, satisfying the canonical commutation relations

$$[\hat{a}_\alpha, \hat{a}_\beta^\dagger] = \delta_{\alpha\beta}, \qquad [\hat{a}_\alpha, \hat{a}_\beta] = [\hat{a}_\alpha^\dagger, \hat{a}_\beta^\dagger] = 0. \tag{2.17}$$

In this case a bosonic coherent state $|a\rangle$ is defined by

$$\hat{a}_\alpha|a\rangle = a_\alpha|a\rangle \tag{2.18}$$

and satisfies

$$|a\rangle = e^{\sum_\alpha a_\alpha \hat{a}_\alpha^\dagger}|0\rangle, \tag{2.19a}$$

$$\langle a|a'\rangle = e^{\sum_\alpha a_\alpha^* a_\alpha'}, \tag{2.19b}$$

$$\int \left( \prod_\alpha \frac{da_\alpha^* da_\alpha}{2\pi i} \right) e^{-\sum_\alpha a_\alpha^* a_\alpha}|a\rangle\langle a| = \hat{1}, \tag{2.19c}$$

$$\text{Tr}\,\hat{Q} = \int \left( \prod_\alpha \frac{da_\alpha^* da_\alpha}{2\pi i} \right) e^{-\sum_\alpha a_\alpha^* a_\alpha}\langle a|\hat{Q}|a\rangle. \tag{2.19d}$$

The path-integral representation of the grand canonical partition function $\mathcal{Z} = \text{Tr}[e^{-\beta(\hat{H}-\mu\hat{N})}]$ is then

$$\mathcal{Z} = \int_{\boldsymbol{a}(\beta\hbar)=\boldsymbol{a}(0)} \mathcal{D}\boldsymbol{a}^*\mathcal{D}\boldsymbol{a}\, e^{-S[\boldsymbol{a}^*,\boldsymbol{a}]/\hbar}, \tag{2.20}$$

where

$$S[\boldsymbol{a}^*,\boldsymbol{a}] = \int_0^{\beta\hbar} d\tau \left[ \sum_\alpha a_\alpha^*(\hbar\partial_\tau - \mu)a_\alpha + H(\boldsymbol{a}^*,\boldsymbol{a}) \right] \tag{2.21}$$

and $\mathcal{D}\boldsymbol{a}^*\mathcal{D}\boldsymbol{a} \equiv \prod_\alpha \mathcal{D}a_\alpha^*\mathcal{D}a_\alpha$. In what follows, we will repeatedly encounter Gaussian path integrals. Accordingly, we will make extensive use of the identity

$$\int \left( \prod_\alpha \frac{da_\alpha^* da_\alpha}{2\pi i} \right) e^{-\sum_{\alpha\beta} a_\alpha^* W_{\alpha\beta} a_\beta} = \frac{1}{\det W}, \tag{2.22}$$

where $W$ is a complex matrix with positive-definite Hermitian part.

## 2.2 Fermionic path integrals

Let us now turn to the case of fermions. Differently from bosons, fermions do not have a correspondence with a classical system. One may therefore wonder how a path-integral description, based on a classical action, can be constructed in this case. Indeed, such a formulation is not possible using ordinary complex numbers. However, by introducing Grassmann numbers, a path-integral description of fermionic systems can be developed in a way that is entirely analogous to the bosonic case.

*Grassmann numbers* are anticommuting quantities defined as the elements of a Grassmann algebra. A Grassmann algebra with $n$ generators $\text{Gr}_n$ is a $\mathbb{C}$-algebra whose generators $\theta_1,\ldots,\theta_n$ satisfy

$$\theta_i\theta_j + \theta_j\theta_i = 0, \qquad i,j = 1,\ldots,n. \tag{2.23}$$

In particular, they are nilpotent: $\theta_i^2 = 0$. As a vector space, $\text{Gr}_n$ has dimension $2^n$, with basis elements given by all distinct monomials formed from the generators, with each generator appearing at most once:

$$\{1, \theta_1, \ldots, \theta_n, \theta_1\theta_2,\ldots,\theta_1\theta_2\theta_3,\ldots\}. \tag{2.24}$$

A general function on this algebra can thus be written as

$$f(\theta_1,\ldots,\theta_n) = \sum_{k=0}^n \sum_{1\le i_1<\cdots<i_k\le n} f_{i_1\ldots i_k}\theta_{i_1}\cdots\theta_{i_k}, \qquad f_{i_1\ldots i_k} \in \mathbb{C}. \tag{2.25}$$

Differentiation with respect to Grassmann numbers can be defined according to the following rules:

$$\frac{\partial\theta_i}{\partial\theta_j} = \delta_{ij}, \qquad \frac{\partial}{\partial\theta_k}(\theta_i\theta_j) = \frac{\partial\theta_i}{\partial\theta_k}\theta_j - \theta_i\frac{\partial\theta_j}{\partial\theta_k} = \delta_{ik}\theta_j - \delta_{jk}\theta_i. \tag{2.26}$$

This means that the derivatives satisfy the anticommutation relations

$$\left\{ \frac{\partial}{\partial\theta_i}, \theta_j \right\} = \delta_{ij}, \qquad \left\{ \frac{\partial}{\partial\theta_i}, \frac{\partial}{\partial\theta_j} \right\} = 0, \tag{2.27}$$

where $\{\hat{A}, \hat{B}\} \equiv \hat{A}\hat{B} + \hat{B}\hat{A}$. Integration over Grassmann numbers is provided by the Berezin integral $\int d\theta$, that is a linear functional satisfying

$$\int d\theta \, [af(\theta) + bg(\theta)] = a \int d\theta \, f(\theta) + b \int d\theta \, g(\theta), \qquad a, b \in \mathbb{C}, \tag{2.28a}$$

$$\int d\theta = 0, \qquad \int d\theta \, \theta = 1, \tag{2.28b}$$

$$\int d\theta_1 \cdots d\theta_n \, \theta_{\sigma(1)} \cdots \theta_{\sigma(n)} = (-1)^{\text{sgn}(\sigma)}, \tag{2.28c}$$

where $\sigma$ is a permutation of $n$ elements and $\text{sgn}(\sigma)$ is its signature. In particular, this integration acts as a differentiation.

For the fermionic path-integral, we need a complex Grassmann algebra with an even number of generators $\theta_1, \overline{\theta}_1, \ldots, \theta_n, \overline{\theta}_n$. In such an algebra there is a natural conjugation operation (an involution) such that

$$\theta_i^* = \overline{\theta}_i, \qquad (\theta_i^*)^* = \theta_i, \qquad (\theta_i \theta_j)^* = \theta_j^* \theta_i^*. \tag{2.29}$$

Using a complex Grassmann algebra, the path integral description of fermionic systems can be given in a manner totally similar to the bosonic case. Consider a Hamiltonian $\hat{H}$ involving a single pair of fermionic creation and annihilation operators $\hat{c}^\dagger$, $\hat{c}$ satisfying the canonical anticommutation relations

$$\{\hat{c}, \hat{c}^\dagger\} = 1, \qquad \{\hat{c}, \hat{c}\} = \{\hat{c}^\dagger, \hat{c}^\dagger\} = 0. \tag{2.30}$$

The Hilbert space has an overcomplete basis constituted by the coherent states $|c\rangle$ defined by

$$\hat{c}|c\rangle = c|c\rangle, \tag{2.31}$$

which satisfy

$$|c\rangle = e^{-c\hat{c}^\dagger}|0\rangle = (1 - c\hat{c}^\dagger)|0\rangle, \tag{2.32a}$$

$$\langle c|c'\rangle = e^{\overline{c}c'} = 1 + \overline{c}c', \tag{2.32b}$$

$$\int d\overline{c} \, dc \, e^{-\overline{c}c} |c\rangle\langle c| = \hat{1}, \tag{2.32c}$$

where $\overline{c}$, $c$ are Grassmann numbers satisfying

$$\{c, c\} = \{\overline{c}, \overline{c}\} = \{c, \overline{c}\} = 0. \tag{2.33}$$

Furthermore, they satisfy

$$\{c, \hat{c}\} = \{\overline{c}, \hat{c}^\dagger\} = \{c, \hat{c}^\dagger\} = 0. \tag{2.34}$$

In fact, the property $\hat{c}^2 = 0$ implies that the variable $c$ defined by Eq. (2.31) must satisfy $c^2 = 0$. The Hermitian conjugate of Eq. (2.31), $\langle c|\hat{c}^\dagger = \overline{c}\langle c|$, requires to introduce another variable $\overline{c}$ which also satisfies $\overline{c}^2 = 0$. In order for the coherent state to be written as in Eq. (2.32a), we also have to impose that $c$ anticommutes with $\hat{c}$; in fact, if $|c\rangle = (1 - c\hat{c}^\dagger)|0\rangle = |0\rangle - c|1\rangle$, then $\hat{c}|c\rangle = -\hat{c}c|1\rangle = +c\hat{c}|1\rangle = c|0\rangle = c(1 - c\hat{c})|0\rangle = c|c\rangle$ only if $c\hat{c} = -\hat{c}c$. Moreover, we have to impose that $c$ anticommutes with $\hat{c}^\dagger$; in fact, $|c\rangle = \{\hat{c}, \hat{c}^\dagger\}|c\rangle = \hat{c}\hat{c}^\dagger(|0\rangle - c|1\rangle) + \hat{c}^\dagger\hat{c}(|0\rangle - c|1\rangle)$ $= \hat{c}|1\rangle + \hat{c}^\dagger c|0\rangle = |0\rangle - c\hat{c}^\dagger|0\rangle = |0\rangle - c|1\rangle = |c\rangle$ only if $c\hat{c}^\dagger = -\hat{c}^\dagger c$. In turn, this implies that the variables $c$ and $\overline{c}$ also anticommute with each other: $c\overline{c} = -\overline{c}c$. This makes $c$ and $\overline{c}$ the generators of the complex Grassmann algebra $\text{Gr}_2$.

As a consequence of the completeness relation (2.32c) and the anticommutation properties of Grassmann numbers, the trace of any operator $\hat{Q} = Q(\hat{c}, \hat{c}^\dagger)$ can be written as

$$\text{Tr}\,\hat{Q} = \int d\bar{c}\, dc\, e^{-\bar{c}c} \langle -c|\hat{Q}|c\rangle. \tag{2.35}$$

Notice that the bra state carries an opposite sign relative to the ket. Following the same steps as in the bosonic case, the canonical partition function is then given by

$$\mathcal{Z} = \text{Tr}\!\left(e^{-\beta \hat{H}}\right) = \int d\bar{c}\, dc\, \langle -c|e^{-\beta \hat{H}}|c\rangle$$

$$= \lim_{M \to \infty} \int \left(\prod_{j=1}^{M} d\bar{c}_j dc_j\right) e^{-\frac{\Delta \tau}{\hbar} \sum_{j=1}^{M}\left[\hbar \bar{c}_j \frac{c_j - c_{j-1}}{\delta \tau} + H(\bar{c}_j, c_{j-1})\right]}, \tag{2.36}$$

with the identification $c = c_M = -c_0$. The exponent in Eq. (2.36) is the discretized version of the Euclidean action

$$S[\bar{c}, c] = \int_0^{\beta \hbar} d\tau \left[\bar{c}(\tau)\hbar \partial_\tau c(\tau) + H(\bar{c}, c)\right], \tag{2.37}$$

where

$$H(\bar{c}, c) \equiv \frac{\langle c|\hat{H}|c\rangle}{\langle c|c\rangle} \tag{2.38}$$

and $\bar{c}(\tau)$, $c(\tau)$ are Grassmann-valued functions, antiperiodic of period $\beta \hbar$. Therefore we will also write

$$\mathcal{Z} = \int_{c(\beta \hbar) = -c(0)} \mathcal{D}\bar{c}\mathcal{D}c\, e^{-S[\bar{c}, c]/\hbar}, \tag{2.39}$$

where

$$\mathcal{D}\bar{c}\mathcal{D}c \equiv \lim_{M \to \infty} \prod_{j=1}^{M} d\bar{c}_j dc_j. \tag{2.40}$$

A many-particle fermionic Hamiltonian involves a complete set of annihilation operators $\{\hat{c}_\alpha\}$ and the corresponding creation operators $\{\hat{c}_\alpha^\dagger\}$, satisfying the canonical anticommutation relations

$$\{\hat{c}_\alpha, \hat{c}_\beta^\dagger\} = \delta_{\alpha\beta}, \qquad \{\hat{c}_\alpha, \hat{c}_\beta\} = \{\hat{c}_\alpha^\dagger, \hat{c}_\beta^\dagger\} = 0. \tag{2.41}$$

In this case a fermionic coherent state $|c\rangle$ is defined by

$$\hat{c}_\alpha|c\rangle = c_\alpha|c\rangle \tag{2.42}$$

and satisfies

$$|c\rangle = e^{-\sum_\alpha c_\alpha \hat{c}_\alpha^\dagger}|0\rangle = \prod_\alpha (1 - c_\alpha \hat{c}_\alpha^\dagger)|0\rangle, \tag{2.43a}$$

$$\langle c|c'\rangle = e^{\sum_\alpha \bar{c}_\alpha c_\alpha'} = \prod_\alpha (1 + \bar{c}_\alpha c_\alpha'), \tag{2.43b}$$

$$\int \left(\prod_\alpha d\bar{c}_\alpha dc_\alpha\right) e^{-\sum_\alpha \bar{c}_\alpha c_\alpha}|c\rangle\langle c| = \hat{1}, \tag{2.43c}$$

$$\text{Tr}\,\hat{Q} = \int \left(\prod_\alpha d\bar{c}_\alpha dc_\alpha\right) e^{-\sum_\alpha \bar{c}_\alpha c_\alpha} \langle -c|\hat{Q}|c\rangle, \tag{2.43d}$$

with

$$\{c_\alpha, c_\beta\} = \{\bar{c}_\alpha, \bar{c}_\beta\} = \{c_\alpha, \bar{c}_\beta\} = 0, \tag{2.44}$$

and

$$\{c_\alpha, \hat{c}_\beta\} = \{\overline{c}_\alpha, \hat{c}_\beta^\dagger\} = \{c_\alpha, \hat{c}_\beta^\dagger\} = 0. \tag{2.45}$$

The path-integral representation of the grand canonical partition function $\mathcal{Z} = \text{Tr}[e^{-\beta(\hat{H}-\mu\hat{N})}]$ is then

$$\mathcal{Z} = \int_{c(\beta\hbar)=-c(0)} \mathcal{D}\overline{c}\mathcal{D}c \, e^{-S[\overline{c},c]/\hbar}, \tag{2.46}$$

where

$$S[\overline{c}, c] = \int_0^{\beta\hbar} d\tau \left[ \sum_\alpha \overline{c}_\alpha(\hbar\partial_\tau - \mu)c_\alpha + H(\overline{c}, c) \right] \tag{2.47}$$

and $\mathcal{D}\overline{c}\mathcal{D}c \equiv \prod_\alpha \mathcal{D}\overline{c}_\alpha \mathcal{D}c_\alpha$. In what follows, we will repeatedly encounter Gaussian path integrals. Accordingly, we will make extensive use of the identity

$$\int \left( \prod_\alpha d\overline{c}_\alpha dc_\alpha \right) e^{-\sum_{\alpha\beta} \overline{c}_\alpha W_{\alpha\beta} c_\beta} = \det W, \tag{2.48}$$

where $W$ is an arbitrary complex matrix.

# 3 Path integrals in imaginary time

Having reviewed the construction of the path integral, we now turn to its application, beginning with the simplest systems that admit a coherent-state path-integral representation: the bosonic and fermionic harmonic oscillators. Despite their simplicity, they capture all essential aspects of the formalism (discretization in imaginary time, boundary conditions, and the continuum limit) and therefore provide a natural starting point for examining the correspondence between the path integral and canonical approaches in a fully controlled setting. Later, in Sections 3.3 and 3.4, we will extend the discussion to the simplest interacting models, the single-site Bose-Hubbard and Hubbard models, and show how their partition functions can be computed exactly in the continuum using the Hubbard-Stratonovich transformation.

## 3.1 Bosonic oscillator

Consider the bosonic oscillator

$$\hat{H} = \varepsilon\hat{a}^\dagger\hat{a} = \varepsilon\hat{N}, \qquad \varepsilon = \hbar\omega. \tag{3.1}$$

For convenience, here we are neglecting the constant zero-point energy $\varepsilon/2$, since it only contributes to the partition function with the term $e^{-\beta\varepsilon/2}$ multiplying the partition function for the Hamiltonian (3.1).

*Hamiltonian approach*—We can directly evaluate the canonical partition function as a trace over the basis of normalized eigenstates $|N\rangle$ of the number operator $\hat{N}$:

$$\mathcal{Z} = \sum_{N=0}^\infty \langle N|e^{-\beta\varepsilon\hat{N}}|N\rangle = \sum_{N=0}^\infty e^{-\beta\varepsilon N} = \frac{1}{1-e^{-\beta\varepsilon}}. \tag{3.2}$$

*Discretized path integral*—In the discretized path integral, we have $H(a_j^*, a_{j-1}) = \varepsilon a_j^* a_{j-1}$ with $a_M = a_0$, and thus at the exponent of Eq. (2.11)

$$-\sum_{j=1}^M \left[ a_j^*(a_j - a_{j-1}) + \tilde{\varepsilon}a_j^* a_{j-1} \right] = -\sum_{j,k=1}^M a_j^*(-\mathcal{G}^{-1})_{jk} a_k, \tag{3.3}$$

where $\tilde{\varepsilon} = \beta \varepsilon / M$ and the matrix $-\mathcal{G}^{-1}$ is given by

$$
-\mathcal{G}^{-1} = \begin{pmatrix}
1 & 0 & 0 & \cdots & 0 & \tilde{\varepsilon} - 1 \\
\tilde{\varepsilon} - 1 & 1 & 0 & \cdots & 0 & 0 \\
0 & \tilde{\varepsilon} - 1 & 1 & \cdots & 0 & 0 \\
\vdots & \vdots & \ddots & \ddots & \vdots & \vdots \\
\vdots & \vdots & \vdots & \ddots & \ddots & \vdots \\
0 & 0 & 0 & \cdots & \tilde{\varepsilon} - 1 & 1
\end{pmatrix}.
\tag{3.4}
$$

The result of the Gaussian integrations over $a_j^*$, $a_j$ is $1/\det(-\mathcal{G}^{-1})$, and the determinant is easily calculated as $1 - (1 - \tilde{\varepsilon})^M$. We thus obtain

$$
\mathcal{Z} = \lim_{M \to \infty} \left[ 1 - \left( 1 - \frac{\beta \varepsilon}{M} \right)^M \right]^{-1} = \frac{1}{1 - e^{-\beta \varepsilon}},
\tag{3.5}
$$

which coincides with the previous result.

**Continuous path integral**—The Euclidean action (2.13) is

$$
S = \int_0^{\beta \hbar} d\tau \, a^*(\tau)(\hbar \partial_\tau + \varepsilon) a(\tau),
\tag{3.6}
$$

hence the partition function (2.15) is

$$
\mathcal{Z} = \int \mathcal{D}a^* \mathcal{D}a \, e^{-\frac{1}{\hbar} \int_0^{\beta \hbar} d\tau \, a^*(\tau)(\hbar \partial_\tau + \varepsilon) a(\tau)}.
\tag{3.7}
$$

Introducing the adimensional variable $u \equiv \tau / \beta \hbar$, this can be rewritten as

$$
\mathcal{Z} = \int \mathcal{D}a^* \mathcal{D}a \, e^{-\int_0^1 du \, a^*(u)(\partial_u + \beta \varepsilon) a(u)} = \det(\partial_u + \beta \varepsilon)^{-1}.
\tag{3.8}
$$

With the periodicity condition used in Eq. (2.15), the functional determinant $\det[\partial_u + f(u)]$, where $f(u)$ is in general a smooth adimensional function of $u$, is given by

$$
\det[\partial_u + f(u)] = 1 - e^{-\int_0^1 du f(u)} = 1 - e^{-\frac{1}{\beta \hbar} \int_0^{\beta \hbar} d\tau f(\tau)}.
\tag{3.9}
$$

To prove this result, we observe that the discretized version of $\int_0^1 du \, a^*(u)[\partial_u + f(u)] a(u)$ is $\sum_{j=1}^M a_j^*(a_j - a_{j-1}) + (f_j/M) a_j^* a_{j-1} = \sum_{j,k=1}^M a_j^* F_{jk} a_k$, where $F_{jk} = \delta_{jk} + (f_j/M - 1)\delta_{j,j-1}$ and $F_{1M} = F_{10} = f_1/M - 1$. The determinant of $F$ is therefore $\det F = 1 - \prod_{j=1}^M (1 - f_j/M)$. Since $f(u)$ is smooth, we can write indifferently $f_j$ or $f_{j-1}$ and exponentiate the product to obtain $\det F = 1 - e^{-\sum_{j=1}^M f_j/M} + O(M^{-2})$, which converges to the functional determinant (3.9) in the continuum limit. In our case, $f(u) = \beta \varepsilon$ is constant and thus we obtain

$$
\mathcal{Z} = \det(\partial_u + \beta \varepsilon)^{-1} = \frac{1}{1 - e^{-\beta \varepsilon}},
\tag{3.10}
$$

which is once again the correct result.

## 3.2 Fermionic oscillator

We consider similarly the fermionic oscillator

$$\hat{H} = \varepsilon \hat{c}^\dagger \hat{c} = \varepsilon \hat{N}. \tag{3.11}$$

*Hamiltonian approach*—The canonical partition function is easily evaluated in the basis of eigenstates of the number operator as

$$\mathcal{Z} = \sum_{N=0,1} \langle N | e^{-\beta \varepsilon \hat{N}} | N \rangle = \sum_{N=0,1} e^{-\beta \varepsilon N} = 1 + e^{-\beta \varepsilon}. \tag{3.12}$$

*Discretized path integral*—In the discretized path integral, we have $H(\overline{c}_j, c_{j-1}) = \varepsilon \overline{c}_j c_{j-1}$ with $c_M = -c_0$, and thus at the exponent of Eq. (2.36)

$$-\sum_{j=1}^{M} \left[ \overline{c}_j (c_j - c_{j-1}) + \tilde{\varepsilon} \overline{c}_j c_{j-1} \right] = -\sum_{j,k=1}^{M} \overline{c}_j (-\mathcal{G}^{-1})_{jk} c_k, \tag{3.13}$$

where $\tilde{\varepsilon} = \beta \varepsilon / M$ and the matrix $-\mathcal{G}^{-1}$ is given by

$$-\mathcal{G}^{-1} = \begin{pmatrix} 1 & 0 & 0 & \cdots & 0 & -(\tilde{\varepsilon}-1) \\ \tilde{\varepsilon}-1 & 1 & 0 & \cdots & 0 & 0 \\ 0 & \tilde{\varepsilon}-1 & 1 & \cdots & 0 & 0 \\ \vdots & \vdots & \ddots & \ddots & \vdots & \vdots \\ \vdots & \vdots & \vdots & \ddots & \ddots & \vdots \\ 0 & 0 & 0 & \cdots & \tilde{\varepsilon}-1 & 1 \end{pmatrix}. \tag{3.14}$$

The result of the Gaussian integrations over $\overline{c}_j$, $c_j$ is $\det(-\mathcal{G}^{-1})$, and the determinant is easily calculated as $1 + (1 - \tilde{\varepsilon})^M$. We thus obtain

$$\mathcal{Z} = \lim_{M \to \infty} \left[ 1 + \left( 1 - \frac{\beta \varepsilon}{M} \right)^M \right] = 1 + e^{-\beta \varepsilon}, \tag{3.15}$$

which coincides with the previous result.

*Continuous path integral*—The Euclidean action (2.37) is

$$S = \int_0^{\beta \hbar} d\tau \, \overline{c}(\tau)(\hbar \partial_\tau + \varepsilon) c(\tau), \tag{3.16}$$

thus the partition function (2.39) is

$$\mathcal{Z} = \int \mathcal{D}\overline{c}\mathcal{D}c \, e^{-\frac{1}{\hbar} \int_0^{\beta \hbar} d\tau \, \overline{c}(\tau)(\hbar \partial_\tau + \varepsilon) c(\tau)}$$

$$= \int \mathcal{D}\overline{c}\mathcal{D}c \, e^{-\int_0^1 du \, \overline{c}(u)(\partial_u + \beta \varepsilon) c(u)} = \det(\partial_u + \beta \varepsilon). \tag{3.17}$$

With the periodicity condition used in Eq. (2.39), the functional determinant $\det[\partial_u + f(u)]$, where $f(u)$ is in general a smooth adimensional function of $u$, is given by

$$\det[\partial_u + f(u)] = 1 + e^{-\int_0^1 du \, f(u)} = 1 + e^{-\frac{1}{\beta \hbar} \int_0^{\beta \hbar} d\tau \, f(\tau)}. \tag{3.18}$$

The proof goes as in the bosonic case. The discretized version of $\int_0^1 du \, \overline{c}(u)[\partial_u + f(u)]c(u)$ is $\sum_{j=1}^{M} \overline{c}_j (c_j - c_{j-1}) + (f_j/M)\overline{c}_j c_{j-1} = \sum_{j,k=1}^{M} \overline{c}_j F_{jk} c_k$, where $F_{jk} = \delta_{jk} + (f_j/M - 1)\delta_{j,j-1}$ and

$F_{1M} = -F_{10} = -(f_1/M - 1)$. The determinant of $F$ is therefore $\det F = 1 + \prod_{j=1}^{M}(1 - f_j/M)$. Since $f(u)$ is smooth, we can write indifferently $f_j$ or $f_{j-1}$ and exponentiate the product to obtain $\det F = 1 + e^{-\sum_{j=1}^{M} f_j/M} + O(M^{-2})$, which converges to the functional determinant (3.18) in the continuum limit. In our case, $f(u) = \beta\varepsilon$ is constant and thus we obtain

$$\mathcal{Z} = \det(\partial_u + \beta\varepsilon) = 1 + e^{-\beta\varepsilon}, \tag{3.19}$$

which is once again the correct result.

## 3.3 Single-site Bose-Hubbard model

We consider as a simple example of interacting theory the single-site Bose-Hubbard model

$$\hat{H} = -\mu\hat{a}^\dagger\hat{a} + \frac{g}{2}\hat{a}^\dagger\hat{a}^\dagger\hat{a}\hat{a} = -\mu\hat{N} + \frac{g}{2}\hat{N}(\hat{N} - 1). \tag{3.20}$$

The canonical partition function is computed in the Hamiltonian approach as

$$\mathcal{Z} = \sum_{N=0}^{\infty} e^{-\beta[-\mu N + \frac{g}{2}N(N-1)]}. \tag{3.21}$$

Let us see how the same result can be obtained with the continuous path integral. The Euclidean action is [Eq. (2.13)]

$$S = \int_0^{\beta\hbar} d\tau \left\{ \frac{1}{2}[a^*(\tau)\dot{a}(\tau) - \dot{a}^*(\tau)a(\tau)] - \mu a^*(\tau)a(\tau) + \frac{g}{2}[a^*(\tau)a(\tau)]^2 \right\}, \tag{3.22}$$

where $\dot{a} \equiv \hbar\partial_\tau a$, and we have integrated by parts the kinetic term to put it in symmetric form. Wilson and Galitski [9] proposed to compute the partition function as follows. Let $a(\tau) = \sqrt{N(\tau)}e^{i\theta(\tau)}$ and $a^*(\tau) = \sqrt{N(\tau)}e^{-i\theta(\tau)}$, so that the measure is $\mathcal{D}a^*\mathcal{D}a = \mathcal{D}N\mathcal{D}\theta$ and the action (3.22) becomes

$$S = \int_0^{\beta\hbar} d\tau \left[ iN(\tau)\dot{\theta}(\tau) + H(N) \right], \tag{3.23}$$

where

$$H(N) = \frac{\langle a|\hat{H}|a\rangle}{\langle a|a\rangle} = -\mu N(\tau) + \frac{g}{2}N(\tau)^2. \tag{3.24}$$

Integrating by parts the term $iN(\tau)\dot{\theta}(\tau)$, we get $S = i\hbar N(0)\Delta\theta + \int_0^{\beta\hbar} d\tau[-i\dot{N}(\tau)\theta(\tau) + H(N)]$, where $N(0) = N(\beta\hbar)$ and $\Delta\theta = \theta(\beta\hbar) - \theta(0) = 2\pi k$. The integer $k$, which counts how many times $\theta$ wraps around the circle as $\tau$ goes from 0 to $\beta\hbar$, defines different topological sectors which contribute to the partition function. The path integral $\int \mathcal{D}\theta\, e^{\frac{i}{\hbar}\int_0^{\beta\hbar} d\tau \dot{N}(\tau)\theta(\tau)}$ then gives $\delta[\dot{N}(\tau)]$, which fixes $N(\tau)$ to the constant $x = N(0) \geq 0$. The partition function is therefore

$$\mathcal{Z} = \sum_{k=-\infty}^{\infty} \int_0^{\infty} dx\, e^{-2\pi ikx} e^{-\beta H(x)}. \tag{3.25}$$

Using the Poisson summation formula[3] $\sum_{k=-\infty}^{\infty} e^{-2\pi ikx} = \sum_{N=-\infty}^{\infty} \delta(x - N)$, and noting that since $x \geq 0$ only non-negative integers contribute, we obtain

$$\mathcal{Z} = \sum_{N=0}^{\infty} e^{-\beta H(N)} = \sum_{N=0}^{\infty} e^{-\beta(-\mu N + \frac{g}{2}N^2)} \qquad \text{(incorrect)}. \tag{3.26}$$

---

[3]The Poisson summation formula is simply the statement that $\sum_{k=-\infty}^{\infty} e^{-2\pi ikx}$ is the Fourier series of the Dirac comb $\Delta(x) = \sum_{N=-\infty}^{\infty} \delta(x - N)$. It is clear that $\Delta(x)$ is periodic with unit period, therefore it can be expanded in Fourier series as $\Delta(x) = \sum_{k=-\infty}^{\infty} \Delta_k e^{-2\pi ikx}$. The Fourier coefficients are $\Delta_k = \int_{-1/2}^{1/2} dx\, \Delta(x)e^{2\pi ikx} = \sum_{N=-\infty}^{\infty} \int_{-1/2}^{1/2} dx\, \delta(x - N)e^{2\pi ikx} = \int_{-1/2}^{1/2} dx\, \delta(x)e^{2\pi ikx} = 1$ for any $k$, which proves the formula.

It is clear from Eq. (3.26) that if $H = \langle a|\hat{H}|a\rangle / \langle a|a\rangle$, written in terms of $N = |a|^2$, is equal to $\langle \hat{H}\rangle_N = \langle N|\hat{H}|N\rangle$, then the result of this path integral calculation is identical to the one obtained from the Hamiltonian approach. This is the case for a Gaussian Hamiltonian, $\hat{H} = \varepsilon \hat{a}^\dagger \hat{a}$, for which $H = \varepsilon N = \langle \hat{H}\rangle_N$. However, it is not the case for interacting Hamiltonians such as (3.20), for which $H = -\mu N + \frac{g}{2}N^2$, whereas $\langle \hat{H}\rangle_N = -\mu N + \frac{g}{2}N(N-1) \neq H$.

Wilson and Galitski deduced from this seemingly exact calculation that the continuous path integral fails to produce the correct result in the cases where the square of $\hat{a}^\dagger \hat{a}$ is involved. This deduction is wrong, because it is based on a mistaken assumption on the continuum limit of the nonlinear change of variables to the number-phase representation. The change of variables is perfectly valid at the level of the discretized path integral, and performing the calculation there does indeed give the correct result. The problem arises in the continuum limit, in assuming that $a^*(\tau)\partial_\tau a(\tau)$ can be replaced by $\frac{1}{2}\partial_\tau N(\tau) + iN(\tau)\partial_\tau \theta(\tau)$, which clearly leads to incorrect results. The correct continuum limit of the number-phase representation was given by Bruckmann and Urbina [15]. We will not discuss their treatment here, but instead present an alternative derivation using the Hubbard-Stratonovich (HS) transformation. As we will see, this too must be handled with care.

### 3.3.1 Hubbard-Stratonovich (HS) transformation

As shown by Rançon [14], the continuous path integral yields the exact result, Eq. (3.21), if all the manipulations are legitimate. The partition function can be computed exactly using a HS transformation, which allows us to decouple the interaction term $H_{int} = \frac{g}{2}[a^*(\tau)a(\tau)]^2$ using the identity

$$e^{-\frac{g}{2\hbar}\int_0^{\beta\hbar} d\tau [a^*(\tau)a(\tau)]^2} = \int \mathcal{D}\phi\, e^{-\frac{1}{\hbar}\int_0^{\beta\hbar} d\tau \left[\frac{1}{2g}\phi(\tau)^2 - i\phi(\tau)a^*(\tau)a(\tau)\right]}, \tag{3.27}$$

where the real HS field $\phi(\tau)$ has the dimensions of energy and the normalization of the Gaussian integral over $\phi(\tau)$ has been included in the measure

$$\mathcal{D}\phi \equiv \lim_{M\to\infty} \prod_{j=1}^{M} \sqrt{\frac{\delta\tau}{2\pi\hbar g}}\, d\phi_j. \tag{3.28}$$

We can thus write the partition function as

$$\mathcal{Z} = \int \mathcal{D}\phi \mathcal{D}a^* \mathcal{D}a\, e^{-S_{HS}[\phi, a^*, a]/\hbar}, \tag{3.29}$$

where

$$S_{HS} = \int_0^{\beta\hbar} d\tau \left\{ \frac{\phi(\tau)^2}{2g} + a^*(\tau)[\hbar\partial_\tau - \mu - i\phi(\tau)]a(\tau) \right\} \tag{3.30}$$

is the HS action. The strategy is now to perform the Gaussian integration over the bosonic fields to obtain an effective action for the HS field, and from this the partition function. If $\phi(\tau)$ were a smooth function, we could apply Eq. (3.9) and compute the partition function as follows:

$$\begin{aligned}
\mathcal{Z} &= \int \mathcal{D}\phi\, \frac{e^{-\frac{1}{\hbar}\int_0^{\beta\hbar} d\tau \frac{1}{2g}\phi(\tau)^2}}{1 - e^{\beta\mu + \frac{i}{\hbar}\int_0^{\beta\hbar} d\tau \phi(\tau)}} \\
&= \sum_{N=0}^{\infty} e^{\beta\mu N} \int \mathcal{D}\phi\, e^{-\frac{1}{\hbar}\int_0^{\beta\hbar} d\tau \left[\frac{1}{2g}\phi(\tau)^2 - iN\phi(\tau)\right]} \\
&= \sum_{N=0}^{\infty} e^{-\beta(-\mu N + \frac{g}{2}N^2)} \qquad \text{(incorrect)},
\end{aligned} \tag{3.31}$$

where in the second line we have used the geometric series identity $(1-x)^{-1} = \sum_{N=0}^{\infty} x^N$. Again we obtain an incorrect result, identical to that in Eq. (3.26). The reason is that $\phi(\tau)$ is not a smooth function, and therefore the functional determinant $\det[\hbar\partial_\tau - \mu - i\phi(\tau)]$ cannot be computed as in Eq. (3.9). In fact, the HS field is governed by the Gaussian action $S_\phi = \int_0^{\beta\hbar} d\tau \frac{1}{2g}\phi(\tau)^2$, which implies that its correlation function diverges at equal times,

$$\int \mathcal{D}\phi \, \phi(\tau)\phi(\tau')e^{-\frac{1}{\hbar}\int_0^{\beta\hbar} d\tau \frac{1}{2g}\phi(\tau)^2} = g\delta\left(\frac{\tau-\tau'}{\hbar}\right). \tag{3.32}$$

The physical origin of this behavior can be understood by looking at the definition (3.27). What we are actually doing is resolving a contact interaction between four $a$ fields in terms of two pairs of $a$ fields exchanging a $\phi$ field; since the original interaction has strength $g$ and is instantaneous, the force-carrying $\phi$ field must satisfy $\langle\hat{\phi}(\tau)\hat{\phi}(\tau')\rangle_\phi = g\delta(\frac{\tau-\tau'}{\hbar})$, that is exactly Eq. (3.32) [7].

This kind of white-noise correlation is responsible for the non-differentiability of $\phi(\tau)$, and calls for a different way to compute functional determinants involving the HS field. The discretized version of Eq. (3.32) is

$$\int \left(\prod_{j=1}^{M} \sqrt{\frac{\delta\tau}{2\pi\hbar g}}d\phi_j\right)\phi_{j_1}\phi_{j_2}e^{-\frac{\delta\tau}{2\hbar g}\sum_{j=1}^{M}\phi_j^2} = g\frac{\delta_{j_1 j_2}}{\delta\tau/\hbar}. \tag{3.33}$$

Since $\phi_j$ is always integrated over at the end, similarly to the noise of a stochastic process, Eq. (3.33) means that we should think of $\phi_j$ as being of order $\delta\tau^{-1/2}$ in all expressions involving it, and all physical quantities are averaged over independent realizations of $\phi_j$ with a Gaussian distribution of variance $\hbar g/\delta\tau$. Therefore, when computing the functional determinant, it is not true that $1 + \frac{\delta\tau}{\hbar}(\mu + i\phi_j) = e^{\frac{\delta\tau}{\hbar}(\mu+i\phi_j)} + O(\delta\tau^2)$, because the expansion of the exponential produces a term proportional to $\delta\tau^2\phi_j^2$, which is actually of order $\delta\tau$, and not of order $\delta\tau^2$ as for smooth functions. One therefore needs to correct the exponentiation for stochastic fields, $1 + \frac{\delta\tau}{\hbar}(\mu + i\phi_j) = e^{\frac{\delta\tau}{\hbar}(\mu+i\phi_j+\frac{\delta\tau}{2\hbar}\phi_j^2)} + O(\delta\tau^2)$, which implies

$$\prod_{j=1}^{M}\left[1 + \frac{\delta\tau}{\hbar}(\mu + i\phi_j)\right] = e^{\frac{\delta\tau}{\hbar}\sum_{j=1}^{M}\left(\mu+i\phi_j+\frac{\delta\tau}{2\hbar}\phi_j^2\right)} + O(\delta\tau^2). \tag{3.34}$$

Although this expression is now correct to order $\delta\tau^2$, it bears the inconvenience that the term $\sum_{j=1}^{M}\frac{1}{2}\left(\frac{\delta\tau}{\hbar}\right)^2\phi_j^2$ does not have a nice continuum limit. Here comes to rescue the observation that since all expressions are to be eventually averaged over $\phi_j$, replacing $\frac{1}{2}(\frac{\delta\tau}{\hbar})^2\phi_j^2$ by $\frac{\delta\tau}{\hbar}\frac{g}{2}$ in all these expressions give a vanishing error in the limit $\delta\tau \to 0$, in a sense that can be made rigorous in the context of Itô calculus, where this is the so-called Itô substitution rule [16]. Therefore, the functional determinant involving the HS field is

$$\det[\partial_u - \beta\mu - i\beta\phi(u)] = 1 - e^{\frac{1}{\hbar}\int_0^{\beta\hbar} d\tau\left[\mu+\frac{g}{2}+i\phi(\tau)\right]}, \tag{3.35}$$

where $u \equiv \tau/\beta\hbar$ as in Eq. (3.9). The correction $g/2$ is similar to a shift of the chemical potential, the origin of which is the stochastic nature of the HS field. The exact partition function is then

$$\mathcal{Z} = \int \mathcal{D}\phi \, e^{-S_{eff}[\phi]/\hbar}, \tag{3.36}$$

where

$$
\begin{aligned}
S_{eff} &= \int_0^{\beta\hbar} d\tau \, \frac{\phi(\tau)^2}{2g} - \hbar \ln\left\{ \int \mathcal{D}a^* \mathcal{D}a \, e^{-\frac{1}{\hbar}\int_0^{\beta\hbar} d\tau \, a^*(\tau)[\hbar\partial_\tau - \mu - i\phi(\tau)]a(\tau)} \right\} \\
&= \int_0^{\beta\hbar} d\tau \, \frac{\phi(\tau)^2}{2g} + \hbar \ln\left\{ 1 - e^{\frac{1}{\hbar}\int_0^{\beta\hbar} d\tau \left[\mu + \frac{g}{2} + i\phi(\tau)\right]} \right\}
\end{aligned}
\tag{3.37}
$$

is the effective action for the HS field. Computing $\mathcal{Z}$ as in Eq. (3.31) now gives the correct result:

$$
\begin{aligned}
\mathcal{Z} &= \int \mathcal{D}\phi \, \frac{e^{-\frac{1}{\hbar}\int_0^{\beta\hbar} d\tau \, \frac{1}{2g}\phi(\tau)^2}}{1 - e^{\beta(\mu+\frac{g}{2}) + \frac{i}{\hbar}\int_0^{\beta\hbar} d\tau \, \phi(\tau)}} \\
&= \sum_{N=0}^\infty e^{\beta(\mu+\frac{g}{2})N} \int \mathcal{D}\phi \, e^{-\frac{1}{\hbar}\int_0^{\beta\hbar} d\tau \left[\frac{1}{2g}\phi(\tau)^2 - iN\phi(\tau)\right]} \\
&= \sum_{N=0}^\infty e^{-\beta\left[-\mu N + \frac{g}{2}N(N-1)\right]}.
\end{aligned}
\tag{3.38}
$$

### 3.3.2   HS transformation and mean-field approximation

We conclude this section by emphasizing the connection between the HS transformation we just discussed and the mean-field approximation. The mean-field approximation for the action $S[a^*, a]$ is based on the assumption that the fluctuations $\delta n(\tau) \equiv a^*(\tau)a(\tau) - N$ of the operator $\hat{a}^\dagger \hat{a}$ around its average $N$ are small, so that we may decouple the interaction term $H_{int} = \frac{g}{2}[a^*(\tau)a(\tau)]^2$ as $H_{int} = \frac{g}{2}[N + \delta n(\tau)]^2 = gNa^*(\tau)a(\tau) - \frac{g}{2}N^2 + O(\delta n^2)$. The mean-field action is therefore

$$
S_{mf}[a^*, a] = -\beta\hbar\frac{g}{2}N^2 + \int_0^{\beta\hbar} d\tau \, a^*(\tau)(\hbar\partial_\tau - \mu + gN)a(\tau).
\tag{3.39}
$$

Using Eq. (3.9), we then obtain the mean-field partition function

$$
\mathcal{Z}_{mf} = \int \mathcal{D}a^* \mathcal{D}a \, e^{-S_{mf}[a^*, a]/\hbar} = \frac{e^{\beta\frac{g}{2}N^2}}{1 - e^{\beta(\mu - gN)}},
\tag{3.40}
$$

where the value of $N$ is fixed by the self-consistency condition

$$
\begin{aligned}
N = \langle \hat{a}^\dagger \hat{a} \rangle_{mf} &= \frac{\int \mathcal{D}a^* \mathcal{D}a \, a^*(\tau)a(\tau)e^{-S_{mf}[a^*, a]/\hbar}}{\int \mathcal{D}a^* \mathcal{D}a \, e^{-S_{mf}[a^*, a]/\hbar}} \\
&= \frac{1}{\beta}\frac{\partial \ln \mathcal{Z}_{mf}}{\partial \mu} \\
&= \frac{1}{e^{\beta(-\mu + gN)} - 1}.
\end{aligned}
\tag{3.41}
$$

Notice that $S_{mf}[a^*, a]$ is, up to a constant, the action of an harmonic oscillator with energy $\varepsilon = -\mu + gN$; consistently, we find that $N$ follows the Bose-Einstein distribution $(e^{\beta\varepsilon} - 1)^{-1}$.

This mean-field approximation is equivalent to a static saddle-point approximation of the HS action, in which one replaces the HS field by its average saddle configuration. In fact, replacing $\phi(\tau) \rightarrow \Phi$ in Eq. (3.30) we obtain

$$
S_{HS\text{-}sp}[a^*, a] = \beta\hbar\frac{\Phi^2}{2g} + \int_0^{\beta\hbar} d\tau \, a^*(\tau)(\hbar\partial_\tau - \mu - i\Phi)a(\tau),
\tag{3.42}
$$

and the correspondence with Eq. (3.39) is given by the identification of $\Phi$ with the average of the saddle configuration $\phi_{sp}(\tau)$ solving $0 = \partial S_{HS}/\partial \phi(\tau)|_{\phi=\phi_{sp}} = \phi_{sp}(\tau) - iga^*(\tau)a(\tau)$:[4]

$$\Phi \equiv \langle \phi_{sp}(\tau) \rangle = ig\langle a^*(\tau)a(\tau) \rangle = igN, \tag{3.43}$$

where $\langle \cdots \rangle$ is defined self-consistently as the average computed using the action $S_{HS\text{-}sp}$ (or $S_{mf}$) itself, as in Eq. (3.41).

We remark that a different approximation of the partition function can be obtained from the saddle-point approximation of the effective action (3.37) for the HS field. In this case $\mathcal{Z}_{sp} = e^{-S_{eff}[\varphi]/\hbar}$, where $\varphi(\tau)$ is the solution of

$$0 = \frac{\partial S_{eff}}{\partial \phi(\tau)}\bigg|_{\phi=\varphi} = \frac{\varphi(\tau)}{g} - \frac{i}{e^{-\frac{1}{\hbar}\int_0^{\beta\hbar} d\tau[\mu+\frac{g}{2}+i\varphi(\tau)]} - 1}. \tag{3.44}$$

This fixes $\varphi$ to the constant

$$\varphi = \frac{ig}{e^{\beta(-\mu-\frac{g}{2}-i\varphi)} - 1}, \tag{3.45}$$

and

$$\mathcal{Z}_{sp} = \frac{e^{-\beta\frac{\varphi^2}{2g}}}{1 - e^{\beta(\mu+\frac{g}{2}+i\varphi)}}. \tag{3.46}$$

We see that $\mathcal{Z}_{sp}$ differs from $\mathcal{Z}_{mf}$ by the effective shift of $g/2$ of the chemical potential, which is a consequence of the fact that in the former case we perform the saddle point approximation *after* computing the path integral over the bosonic fields.

## 3.4 Single-site Hubbard model

A fermionic analogue of the single-site Bose-Hubbard model discussed in Section 3.3 is the single-site Hubbard model for spin-$\frac{1}{2}$ fermions,

$$\hat{H} = \sum_{\sigma=\uparrow,\downarrow} \varepsilon\hat{c}_\sigma^\dagger\hat{c}_\sigma + g\hat{c}_\uparrow^\dagger\hat{c}_\downarrow^\dagger\hat{c}_\downarrow\hat{c}_\uparrow = \varepsilon(\hat{N}_\uparrow + \hat{N}_\downarrow) + g\hat{N}_\uparrow\hat{N}_\downarrow, \tag{3.47}$$

whose partition function is readily evaluated in the basis of eigenstates $|N_\uparrow, N_\downarrow\rangle$, with $N_\sigma = 0, 1$, $\sigma = \uparrow, \downarrow$, as

$$\mathcal{Z} = 1 + 2e^{-\beta\varepsilon} + e^{-\beta(2\varepsilon+g)}. \tag{3.48}$$

Now consider the continuous path integral. The corresponding Euclidean action is [Eq. (2.37)]

$$S = \int_0^{\beta\hbar} d\tau \left[ \sum_{\sigma=\uparrow,\downarrow} \bar{c}_\sigma(\tau)(\hbar\partial_\tau + \varepsilon)c_\sigma(\tau) + g\bar{c}_\uparrow(\tau)\bar{c}_\downarrow(\tau)c_\downarrow(\tau)c_\uparrow(\tau) \right]. \tag{3.49}$$

The partition function can be computed exactly using a HS transformation, which allows us to decouple the interaction term $H_{int} = g\bar{c}_\uparrow(\tau)\bar{c}_\downarrow(\tau)c_\downarrow(\tau)c_\uparrow(\tau)$ using the identity

$$e^{-\frac{g}{\hbar}\int_0^{\beta\hbar} d\tau\,\bar{c}_\uparrow(\tau)\bar{c}_\downarrow(\tau)c_\downarrow(\tau)c_\uparrow(\tau)} = \int \mathcal{D}\phi\, e^{-\frac{1}{\hbar}\int_0^{\beta\hbar} d\tau\left[\frac{\phi_1(\tau)\phi_2(\tau)}{g} - i\phi_1(\tau)\bar{c}_\uparrow(\tau)c_\uparrow(\tau) - i\phi_2(\tau)\bar{c}_\downarrow(\tau)c_\downarrow(\tau)\right]}, \tag{3.50}$$

---

[4]This implies that the saddle configuration $\phi_{sp}(\tau)$ is purely imaginary. Since $S_{HS}$ is holomorphic, we can deform the real integration contour over $\phi$ of Eq. (3.30) in the complex $\phi$ plane without changing the value of $\mathcal{Z}$. The physically relevant saddle that dominates the integral is purely imaginary.

where the measure $\mathcal{D}\phi \equiv \mathcal{D}\phi_1 \mathcal{D}\phi_2$ is normalized so that $\int \mathcal{D}\phi\, e^{-\frac{1}{\hbar}\int_0^{\beta\hbar} d\tau \frac{\phi_1(\tau)\phi_2(\tau)}{g}} = 1$ [17]. We can thus write the partition function as

$$\mathcal{Z} = \int \mathcal{D}\phi\, \mathcal{D}\bar{c}_\sigma\, \mathcal{D}c_\sigma\, e^{-S_{HS}[\phi_1,\phi_2,\bar{c}_\sigma,c_\sigma]/\hbar}, \tag{3.51}$$

where

$$S_{HS} = \int_0^{\beta\hbar} d\tau \left\{ \frac{\phi_1(\tau)\phi_2(\tau)}{g} + \sum_{\sigma=\uparrow(1),\downarrow(2)} \bar{c}_\sigma(\tau)[\hbar\partial_\tau + \varepsilon - i\phi_\sigma(\tau)]c_\sigma(\tau) \right\}. \tag{3.52}$$

Here we observe an important distinction relative to the bosonic case. Now we have *two* HS fields, governed by the action $S_\phi = \int_0^{\beta\hbar} d\tau\, \frac{1}{g}\phi_1(\tau)\phi_2(\tau)$. This structure permits only instantaneous $\phi_1$-$\phi_2$ mixing, which mediates the interaction between the four $c$ fields, while neither $\phi_1$ nor $\phi_2$ propagate independently. That is, the correlation functions are

$$\langle \hat{\phi}_1(\tau)\hat{\phi}_2(\tau') \rangle_\phi = g\delta\left(\frac{\tau-\tau'}{\hbar}\right), \qquad \langle \hat{\phi}_1(\tau)\hat{\phi}_1(\tau') \rangle_\phi = \langle \hat{\phi}_2(\tau)\hat{\phi}_2(\tau') \rangle_\phi = 0. \tag{3.53}$$

The two Gaussian integrations over the fermionic fields in Eq. (3.51) yield the product of two independent functional determinants, $\det[\hbar\partial_\tau + \varepsilon - i\phi_1(\tau)]\det[\hbar\partial_\tau + \varepsilon - i\phi_2(\tau)]$. Since the two HS fields are not auto-correlated, they can be treated as smooth functions, so that we can compute the functional determinants according to Eq. (3.18). Hence, no Itô correction is needed in the present case. The result for the partition function is indeed

$$\mathcal{Z} = \int \mathcal{D}\phi \left\{ 1 + e^{-\frac{1}{\hbar}\int_0^{\beta\hbar} d\tau[\varepsilon - i\phi_1(\tau)]} \right\} \left\{ 1 + e^{-\frac{1}{\hbar}\int_0^{\beta\hbar} d\tau[\varepsilon - i\phi_2(\tau)]} \right\} e^{-\frac{1}{\hbar}\int_0^{\beta\hbar} d\tau \frac{\phi_1(\tau)\phi_2(\tau)}{g}}$$

$$= \int \mathcal{D}\phi \left\{ 1 + e^{-\beta\varepsilon}\left[ e^{\frac{i}{\hbar}\int_0^{\beta\hbar} d\tau\, \phi_1(\tau)} + e^{\frac{i}{\hbar}\int_0^{\beta\hbar} d\tau\, \phi_2(\tau)} \right] \right.$$

$$\left. + e^{-2\beta\varepsilon} e^{\frac{i}{\hbar}\int_0^{\beta\hbar} d\tau[\phi_1(\tau)+\phi_2(\tau)]} \right\} e^{-\frac{1}{\hbar}\int_0^{\beta\hbar} d\tau \frac{\phi_1(\tau)\phi_2(\tau)}{g}}$$

$$= 1 + 2e^{-\beta\varepsilon} \int \mathcal{D}\phi\, e^{-\frac{1}{\hbar}\int_0^{\beta\hbar} d\tau \frac{\phi_1(\tau)}{g}[\phi_2(\tau)-ig]}$$

$$+ e^{-2\beta\varepsilon} \int \mathcal{D}\phi\, e^{-\frac{1}{\hbar}\int_0^{\beta\hbar} d\tau \left\{ \frac{\phi_1(\tau)\phi_2(\tau)}{g} - i[\phi_1(\tau)+\phi_2(\tau)] \right\}}$$

$$= 1 + 2e^{-\beta\varepsilon} + e^{-2\beta\varepsilon} e^{-\beta g}, \tag{3.54}$$

which coincides with Eq. (3.48).

# 4 Path integrals in frequency space

In most applications, coherent-state path integrals are typically evaluated in frequency space. In fact, since $a(\tau)$ and $c(\tau)$ (and their conjugates) are, respectively, periodic and antiperiodic with period $\beta\hbar$, we may expand them in Fourier series with respect to bosonic and fermionic Matsubara frequencies, defined by

$$\omega_n = \begin{cases} \frac{2\pi n}{\beta\hbar} & \text{bosons}, \\ \frac{(2n+1)\pi}{\beta\hbar} & \text{fermions}, \end{cases} \qquad n \in \mathbb{Z}. \tag{4.1}$$

However, in doing this we must be careful, as it is important to remember that by construction, time-ordering is implicit in the path integral. This means that $a^*(\tau)$ and $\bar{c}(\tau)$ always appear at a *slightly later* time than $a(\tau)$ and $c(\tau)$ in the action. Introducing the unitary notation $\overline{\alpha}(\tau)$ for $a^*(\tau)$ and $\bar{c}(\tau)$, and $\alpha(\tau)$ for $a(\tau)$ and $c(\tau)$, we should thus replace $\overline{\alpha}(\tau) \to \overline{\alpha}(\tau^+)$, where $\tau^+ \equiv \tau + 0^+$. When expanding with respect to Matsubara frequencies we will then have

$$\overline{\alpha}(\tau^+) = \sum_{n=-\infty}^{\infty} \overline{\alpha}_n e^{i\omega_n(\tau+0^+)}, \qquad \alpha(\tau) = \sum_{n=-\infty}^{\infty} \alpha_n e^{-i\omega_n\tau}. \tag{4.2}$$

Consider for instance a bosonic or fermionic oscillator with $\hat{H} = \hbar\omega\hat{a}^\dagger\hat{a}$. Its Euclidean action is given by $S = \int_0^{\beta\hbar} d\tau\, \overline{\alpha}(\tau^+)(\hbar\partial_\tau + \hbar\omega)\alpha(\tau)$ [Eqs. (3.6) and (3.16)]. According to Eq. (4.2), the precise form of the action in frequency space is

$$S = \beta\hbar \sum_{n=-\infty}^{\infty} \overline{\alpha}_n(-i\hbar\omega_n + \hbar\omega)\alpha_n e^{i\omega_n 0^+}. \tag{4.3}$$

The transformation from imaginary times to Matsubara frequencies has unit Jacobian, and the partition function is given by the Gaussian integral

$$\begin{aligned}
\mathcal{Z} &= \int \left[ \prod_{n=-\infty}^{\infty} \frac{d\overline{\alpha}_n\, d\alpha_n}{(2\pi i)^{\delta_{\zeta,1}}} \right] e^{-\beta \sum_{n=-\infty}^{\infty} \overline{\alpha}_n(-i\hbar\omega_n+\hbar\omega)\alpha_n e^{i\omega_n 0^+}} \\
&= \prod_{n=-\infty}^{\infty} \int \frac{d\overline{\alpha}_n\, d\alpha_n}{(2\pi i)^{\delta_{\zeta,1}}} e^{-\overline{\alpha}_n[\beta\hbar(-i\omega_n+\omega)e^{i\omega_n 0^+}]\alpha_n} \\
&= \prod_{n=-\infty}^{\infty} \left[ \beta\hbar(-i\omega_n + \omega)e^{i\omega_n 0^+} \right]^{-\zeta},
\end{aligned} \tag{4.4}$$

where

$$\zeta \equiv \begin{cases} +1 & \text{bosons,} \\ -1 & \text{fermions.} \end{cases} \tag{4.5}$$

As expected, the infinite product in Eq. (4.4) is real, because the Matsubara frequencies come in $\pm\omega_n$ pairs, and $(-i\omega_n + \omega)(i\omega_n + \omega) = \omega_n^2 + \omega^2 \in \mathbb{R}$. The natural logarithm of $\mathcal{Z}$ is then

$$\begin{aligned}
\ln\mathcal{Z} &= -\zeta \sum_{n=-\infty}^{\infty} \ln\left[ \beta\hbar(-i\omega_n + \omega)e^{i\omega_n 0^+} \right] \\
&= -\zeta \sum_{n=-\infty}^{\infty} \ln\left[ \beta\hbar(-i\omega_n + \omega) \right] e^{i\omega_n 0^+}.
\end{aligned} \tag{4.6}$$

In the second line we have used the fact that $\delta$ is infinitesimal to replace an expression of the form $\ln(f e^{i\omega_n\delta}) = \ln f + i\omega_n\delta$ with the expression $(\ln f)e^{i\omega_n\delta} = \ln f + i\omega_n\delta \ln f + O(\delta^2)$, a substitution which is valid in the limit $\delta \to 0^+$. The additional $e^{i\omega_n 0^+}$ in Eq. (4.6) serves as a convergence factor that regularizes otherwise ill-convergent Matsubara frequency summations. The time-ordering of the path integral is thus reflected in the prescription that when performing calculations in the Matsubara frequency representation, we should include a convergence factor $e^{i\omega_n\delta}$, with $\delta > 0$, and eventually take the limit $\delta \to 0^+$ at the end of the calculations [4–6]. The summation in Eq. (4.6) then gives

$$-\zeta \ln\mathcal{Z} = \lim_{\delta\to 0^+} \sum_{n=-\infty}^{\infty} \ln[\beta\hbar(-i\omega_n + \omega)]e^{i\omega_n\delta} = \ln\left(1 - \zeta e^{-\beta\hbar\omega}\right), \tag{4.7}$$

which we know to be the exact result, see Eqs. (3.2) and (3.12).

This crucial result can be proved in several ways, typically by making use of techniques of finite-temperature field theory based on complex integration, which we review in the following section. Before that, let us present an alternative approach based on the representation of the logarithm as the Frullani integral $\ln a = \int_0^\infty ds\,(e^{-s} - e^{-as})/s$. This can be written equivalently as

$$\ln a = -\gamma - \Pf_{\delta\to 0^+} \int_\delta^\infty \frac{ds}{s}\, e^{-as}, \tag{4.8}$$

where $\gamma$ is the Euler-Mascheroni constant and $\Pf_{\delta\to 0^+}$ denotes the finite part of the integral in the limit $\delta \to 0^+$. Up to an unimportant numerical constant, we can thus write

$$-\zeta \ln \mathcal{Z} = \sum_{n=-\infty}^\infty \ln[\beta\hbar(-i\omega_n + \omega)] = -\Pf_{\delta\to 0^+} \int_\delta^\infty \frac{ds}{s}\, \zeta^s e^{-\beta\hbar\omega s} \sum_{n=-\infty}^\infty e^{2\pi i n s}. \tag{4.9}$$

Using the Poisson summation formula $\sum_{n=-\infty}^\infty e^{2\pi i n s} = \sum_{n=-\infty}^\infty \delta(s-n)$, we obtain indeed

$$-\zeta \ln \mathcal{Z} = -\Pf_{\delta\to 0^+} \int_\epsilon^\infty \frac{ds}{s}\, \zeta^s e^{-\beta\hbar\omega s} \sum_{n=-\infty}^\infty \delta(s-n)$$
$$= -\sum_{n=1}^\infty \frac{(\zeta e^{-\beta\hbar\omega})^n}{n} = \ln\big(1 - \zeta e^{-\beta\hbar\omega}\big). \tag{4.10}$$

## 4.1 Summations of Matsubara frequencies

The standard scheme to perform summations such as that in Eq. (4.7) is based on the residue theorem and the fact that the functions $n_\zeta(z) = (e^{\beta\hbar z} - \zeta)^{-1}$, the extensions of the Bose and Fermi distributions to the complex $z$ plane, have poles with residue $(\zeta\beta\hbar)^{-1}$ in $z = i\omega_n$ [4–7]:

$$\Res_{z=i\omega_n} \frac{1}{e^{\beta\hbar z} - \zeta} = \lim_{z\to i\omega_n} \frac{z - i\omega_n}{e^{\beta\hbar z} - \zeta} = \lim_{z\to i\omega_n} \frac{z - i\omega_n}{e^{\beta\hbar i\omega_n} e^{\beta\hbar(z-i\omega_n)} - \zeta}$$
$$= \lim_{z\to i\omega_n} \frac{z - i\omega_n}{\zeta[e^{\beta\hbar(z-i\omega_n)} - 1]} = \frac{1}{\zeta\beta\hbar}. \tag{4.11}$$

For any function $f(z)$ holomorphic in $z = i\omega_n$, these facts allow us to write

$$\sum_{\omega_n} f(i\omega_n) = \zeta\beta\hbar \sum_{\omega_n} \Res_{z=i\omega_n} \big[n_\zeta(z)f(z)\big] = \zeta\beta\hbar \oint_\mathcal{C} \frac{dz}{2\pi i}\, n_\zeta(z)f(z), \tag{4.12}$$

where $\mathcal{C}$ is a positively-oriented contour that fully encloses the imaginary axis. This contour integral is usually intractable; however, as long as we are careful not to cross any singularity of $n_\zeta(z)$ and any singularity or branch cut of $f(z)$, Cauchy's integral theorem allows us to deform the integration path to a contour along which the integral can actually be performed. In particular, if $n_\zeta(z)f(z)$ decays faster than $|z|^{-1}$, i.e. $|z||n_\zeta(z)f(z)| \ll 1$ for $|z| \to \infty$, we can inflate the original contour to an infinitely large circle. The integral along the outer perimeter of the contour then vanishes and we are left with the integral along a negatively-oriented contour $\Gamma$ around the branch cuts and the singularities $z_k$ of $f(z)$, so that

$$\sum_{\omega_n} f(i\omega_n) = \zeta\beta\hbar \oint_\Gamma \frac{dz}{2\pi i}\, n_\zeta(z)f(z) = -\zeta\beta\hbar \sum_{z_k} \Res_{z=z_k} \big[n_\zeta(z)f(z)\big], \tag{4.13}$$

where the final equality holds when $f(z)$ has only a discrete number of singularities.

In the case at hand, $f(z) = \ln[\beta\hbar(-z + \omega)]e^{\delta z}$; we have that $|n_\zeta(z)f(z)|$ behaves like $e^{-(\beta\hbar-\delta)\text{Re}(z)}|\ln(-\beta\hbar z)|$ for $\text{Re}(z) \to \infty$ and like $e^{-\delta|\text{Re}(z)|}|\ln(-\beta\hbar z)|$ for $\text{Re}(z) \to -\infty$. Thus

for any $0 < \delta < \beta\hbar$ the integrand is exponentially suppressed at infinity, and we can inflate $\mathcal{C}$ to an infinitely large circle avoiding the branch cut on the positive real axis for $x = \mathrm{Re}(z) \geq \omega > 0$, where $f(x + i0^+) - f(x - i0^+) = -2\pi i$. Then by Eq. (4.13), taking the limit $\delta \to 0^+$,

$$
\begin{aligned}
-\zeta \ln \mathcal{Z} &= \zeta\beta\hbar \int_\omega^\infty \frac{dx}{2\pi i} \frac{f(x + i0^+) - f(x - i0^+)}{e^{\beta\hbar x} - \zeta} \\
&= -\zeta\beta\hbar \int_\omega^\infty \frac{dx}{e^{\beta\hbar x} - \zeta} = \ln\left(1 - \zeta e^{-\beta\hbar\omega}\right),
\end{aligned}
\tag{4.14}
$$

which proves Eq. (4.7).

Another possibility is to differentiate $-\zeta \ln \mathcal{Z}$ with respect to $\omega$, obtaining

$$
\frac{\partial(-\zeta \ln \mathcal{Z})}{\partial \omega} = \lim_{\delta \to 0^+} \sum_{n=-\infty}^\infty \frac{e^{i\omega_n \delta}}{-i\omega_n + \omega}.
\tag{4.15}
$$

In this case, $f(z) = e^{\delta z}/(-z + \omega)$, and $|n_\zeta(z)f(z)|$ behaves like $e^{-(\beta\hbar - \delta)\mathrm{Re}(z)}/|z|$ for $\mathrm{Re}(z) \to \infty$ and like $e^{-\delta|\mathrm{Re}(z)|}/|z|$ for $\mathrm{Re}(z) \to -\infty$. Thus, for any $0 < \delta < \beta\hbar$ the integrand is exponentially suppressed at infinity, and we can inflate $\mathcal{C}$ to an infinitely large circle avoiding the single pole in $z = \omega$. Then by Eq. (4.13), taking the limit $\delta \to 0^+$,

$$
\frac{\partial(-\zeta \ln \mathcal{Z})}{\partial \omega} = -\zeta\beta\hbar \operatorname*{Res}_{z=\omega} \frac{1}{(-z + \omega)(e^{\beta\hbar z} - \zeta)} = \frac{\zeta\beta\hbar}{e^{\beta\hbar\omega} - \zeta}.
\tag{4.16}
$$

Integrating we get $-\zeta \ln \mathcal{Z} = \ln(1 - \zeta e^{-\beta\hbar\omega}) + \mathcal{N}$, where $\mathcal{N}$ is a constant independent of $\omega$. Since $\mathcal{N}$ is dimensionless, it cannot depend on $\beta$ alone and is therefore a pure number. Taking the limit $\beta \to \infty$, where we know that $\ln \mathcal{Z} \to 0$, unambiguously sets $\mathcal{N}$ to zero.

***The necessity for the convergence factor***—This second approach illustrates well the importance of the convergence factor $e^{i\omega_n 0^+}$. Suppose we neglect this factor, that is, we forget the implicit time-ordering of the path integral, and simply consider

$$
\sum_{n=-\infty}^\infty \frac{1}{-i\omega_n + \omega}.
\tag{4.17}
$$

Here $f(z) = 1/(-z + \omega)$, and $|n_\zeta(z)f(z)|$ behaves like $e^{-\beta\hbar\mathrm{Re}(z)}/|z|$ for $\mathrm{Re}(z) \to \infty$ and like $1/|z|$ for $\mathrm{Re}(z) \to -\infty$. The integrand is exponentially suppressed at infinity in the right half-plane, but it only decays as $|z|^{-1}$ in the left half-plane. Therefore, if we inflate $\mathcal{C}$ to an infinitely large circle avoiding the single pole in $z = \omega$, we will have the contribution of the negatively-oriented integral around $\omega$ *and*, in addition, the contribution from the positively-oriented integral along the half circle in the left half-plane. The latter is given by

$$
\zeta\beta\hbar \int_{\text{half circle}} \frac{dz}{2\pi i} \frac{1}{\zeta z} = \beta\hbar \int_{\pi/2}^{3\pi/2} \frac{d\theta}{2\pi} = \frac{\beta\hbar}{2},
\tag{4.18}
$$

therefore

$$
\sum_{n=-\infty}^\infty \frac{1}{-i\omega_n + \omega} = \frac{\beta\hbar}{2} + \frac{\zeta\beta\hbar}{e^{\beta\hbar\omega} - \zeta}.
\tag{4.19}
$$

This is *not* $\partial(-\zeta \ln \mathcal{Z})/\partial \omega$, since it would imply that $-\zeta \ln \mathcal{Z} = \beta\hbar\omega/2 + \ln(1 - \zeta e^{-\beta\hbar\omega})$, which is an incorrect result [see also the discussion following Eq. (4.24)]. We thus see that the presence of the factor $e^{i\omega_n 0^+}$ is necessary to properly regularize $\partial(-\zeta \ln \mathcal{Z})/\partial \omega$ so as to obtain the correct expression for the partition function.

## 4.2 Summations of Matsubara frequencies when the action is in matrix form

Matsubara frequencies possess a simple parity property. Defining

$$
\omega_\ell = \frac{\pi \ell}{\beta \hbar}, \qquad \text{with} \qquad \ell = \begin{cases} 2n & \text{bosons,} \\ 2n+1 & \text{fermions,} \end{cases} \qquad n \in \mathbb{Z}, \qquad (4.20)
$$

it follows immediately that

$$
\omega_{-\ell} = -\omega_\ell. \qquad (4.21)
$$

For example, this allows us to rewrite Eq. (4.6) as

$$
\begin{aligned}
-\zeta \ln \mathcal{Z} &= \sum_\ell \ln \left[ \beta \hbar (-i\omega_\ell + \omega) e^{i\omega_\ell 0^+} \right] \\
&= \delta_{\zeta,1} \ln(\beta \hbar \omega) + \sum_{\ell > 0} \left\{ \ln \left[ \beta \hbar (-i\omega_\ell + \omega) e^{i\omega_\ell 0^+} \right] + \ln \left[ \beta \hbar (i\omega_\ell + \omega) e^{-i\omega_\ell 0^+} \right] \right\} \\
&= \delta_{\zeta,1} \frac{1}{2} \ln(\beta^2 \hbar^2 \omega^2) + \frac{1}{2} \sum_{\ell \neq 0} \ln \left[ \beta^2 \hbar^2 (\omega_\ell^2 + \omega^2) \right] \\
&= \frac{1}{2} \sum_\ell \ln \left[ \beta^2 \hbar^2 (\omega_\ell^2 + \omega^2) \right], \qquad (4.22)
\end{aligned}
$$

where in the second line we used the symmetry $\omega_{-\ell} = -\omega_\ell$, taking care that the term $\omega_\ell = 0$ (in the bosonic case) is the only one not doubled by the symmetry. Here and in the following, it is understood that $\ell$ takes on even (odd) integers in the bosonic (fermionic) case. While this pairing manipulation is algebraically valid term by term for symmetric finite truncations, i.e. for $\ell \in [-N, N]$, the limit $N \to \infty$ is clearly problematic, since the summation in the first line of Eq. (4.22) is convergent, while the summation in the last line is manifestly divergent. A regularization scheme is therefore needed to make sense of Matsubara frequency summations that exploit this symmetry property. More generally, summations of the type encountered in the last line of Eq. (4.22) arise naturally in the evaluation of the partition function as a path integral over multi-component fields, when the action is expressed in matrix form. As we shall see, the appropriate regularization procedure is once again dictated by the underlying construction of the path integral and its associated time-ordering.

Naively, one might hope to be able to avoid regularization by relying on the fact that differentiating Eq. (4.22) with respect to $\omega$ one obtains a finite result,

$$
\frac{\partial(-\zeta \ln \mathcal{Z})}{\partial \omega} = \sum_\ell \frac{\omega}{\omega_\ell^2 + \omega^2} = \begin{cases} \frac{\beta \hbar}{2} \coth\left(\frac{\beta \hbar \omega}{2}\right) & \text{bosons,} \\ \frac{\beta \hbar}{2} \tanh\left(\frac{\beta \hbar \omega}{2}\right) & \text{fermions,} \end{cases} \qquad \text{(incorrect).} \qquad (4.23)
$$

Integrating and setting to $\ln 2$ the arbitrary numerical constant, we get

$$
-\zeta \ln \mathcal{Z} = \frac{\beta \hbar \omega}{2} + \ln\left(1 - \zeta e^{-\beta \hbar \omega}\right) \qquad \text{(incorrect),} \qquad (4.24)
$$

that is the same result following from Eq. (4.19). Despite being presented in several textbooks on finite-temperature field theory, e.g. Refs. [5, 18], this computation leads to an incorrect result. Apparently this problem is not given much importance, perhaps because the spurious term $\beta \hbar \omega / 2$ contributes to the free energy $-\beta^{-1} \ln \mathcal{Z}$ merely as the constant $\zeta \hbar \omega / 2$. This constant energy is then removed *a posteriori*, based on the fact that in the limit $\beta \to \infty$ the free energy must equal the ground state energy of the Hamiltonian, which in the present case is zero. However, this procedure is *ad hoc* and lacks a sound mathematical justification, and thus should definitely be avoided. Crucially, there is a difference between using the known

asymptotic behavior of the free energy to fix an overall numerical constant, as we did after Eq. (4.16), and using it to cancel a dimensional quantity that depends on system parameters, here the frequency $\omega$. While energies may be defined up to an additive constant, the fact that the magnitude of that constant depends on the physical properties of the system is unsettling.

Introducing by hand the usual convergence factor $e^{i\omega_\ell 0^+}$ in Eq. (4.22) does not solve the issue either. In fact, the function $f(z) = \ln[\beta^2\hbar^2(-z^2 + \omega^2)]e^{\delta z}$ has two branch cuts, one on the positive real axis for $x = \text{Re}(z) \geq \omega$, and one on the negative real axis for $x \leq -\omega$, where $f(x + i0^+) - f(x - i0^+) = -\text{sgn}(x)2\pi i$. Therefore

$$
\lim_{\delta \to 0^+} \frac{1}{2} \sum_\ell \ln\left[\beta^2\hbar^2(\omega_\ell^2 + \omega^2)\right]e^{i\omega_n\delta}
$$

$$
= \lim_{\delta \to 0^+} \frac{\zeta\beta\hbar}{2}\left[\int_\omega^\infty \frac{dx}{2\pi i}\frac{(-2\pi i)e^{\delta x}}{e^{\beta\hbar x} - \zeta} + \int_{-\infty}^{-\omega} \frac{dx}{2\pi i}\frac{(2\pi i)e^{\delta x}}{e^{\beta\hbar x} - \zeta}\right]
$$

$$
= -\frac{\zeta\beta\hbar}{2}\left[\int_\omega^\infty \frac{dx}{e^{\beta\hbar x} - 1} + \lim_{\delta \to 0^+}\int_\omega^\infty dx\, \frac{e^{-\delta x}}{\zeta - e^{-\beta\hbar x}}\right]. \tag{4.25}
$$

The first integral, which we already evaluated in Eq. (4.14), yields $-(\zeta/\beta\hbar)\ln(1 - \zeta e^{-\beta\hbar\omega})$. The second integral can be evaluated as follows:

$$
\int_\omega^\infty dx\, \frac{e^{-\delta x}}{\zeta - e^{-\beta\hbar x}} = \zeta\int_\omega^\infty dx\, \frac{e^{-\delta x}}{1 - \zeta e^{-\beta\hbar x}}
$$

$$
= \zeta\sum_{n=0}^\infty \int_\omega^\infty dx\, \zeta^n e^{-(\delta + \beta\hbar n)x}
$$

$$
= \zeta\sum_{n=0}^\infty \zeta^n \frac{e^{-(\delta + \beta\hbar n)\omega}}{\delta + \beta\hbar n}
$$

$$
= \zeta\left[\frac{e^{-\delta\omega}}{\delta} + \sum_{n=1}^\infty \zeta^n \frac{e^{-(\delta + \beta\hbar n)\omega}}{\delta + \beta\hbar n}\right], \tag{4.26}
$$

therefore in the limit of small $\delta$,

$$
\int_\omega^\infty dx\, \frac{e^{-\delta x}}{\zeta - e^{-\beta\hbar x}} = \zeta\left[\frac{1}{\delta} - \omega + \frac{1}{\beta\hbar}\sum_{n=1}^\infty \frac{(\zeta e^{-\beta\hbar\omega})^n}{n}\right] + O(\delta^2). \tag{4.27}
$$

Taking the finite part of the integral, and using result (4.10) for the summation, we thus obtain

$$
\underset{\delta \to 0^+}{\text{Pf}}\int_\omega^\infty dx\, \frac{e^{-\delta x}}{\zeta - e^{-\beta\hbar x}} = \zeta\left[-\omega - \frac{1}{\beta\hbar}\ln\left(1 - \zeta e^{-\beta\hbar\omega}\right)\right]. \tag{4.28}
$$

Putting things together, we arrive at the conclusion that

$$
\lim_{\delta \to 0^+} \frac{1}{2}\sum_\ell \ln\left[\beta^2\hbar^2(\omega_\ell^2 + \omega^2)\right]e^{i\omega_\ell\delta} = \frac{\beta\hbar\omega}{2} + \ln\left(1 - \zeta e^{-\beta\hbar\omega}\right), \tag{4.29}
$$

with the term $\beta\hbar\omega/2$ coming from the integration around the branch cut in the left-half plane.

The reason for this apparent difficulty is that, as anticipated, Eq. (4.22) corresponds to an action written in matrix form, and different components of the matrix require different convergence factors due to the different time-ordering [6]. This can be clearly seen by writing

$$S = \int_0^{\beta\hbar} d\tau \, \overline{\alpha}(\tau^+)(\hbar\partial_\tau + \hbar\omega)\alpha(\tau)$$

$$= \frac{1}{2} \int_0^{\beta\hbar} d\tau \left(\overline{\alpha}(\tau^+) \quad \alpha(\tau)\right) \begin{pmatrix} \hbar\partial_\tau + \hbar\omega & 0 \\ 0 & \zeta(-\hbar\partial_\tau + \hbar\omega) \end{pmatrix} \begin{pmatrix} \alpha(\tau) \\ \overline{\alpha}(\tau^+) \end{pmatrix}$$

$$= \frac{\beta\hbar}{2} \sum_\ell \left(\overline{\alpha}_\ell \quad \alpha_{-\ell}\right) \begin{pmatrix} (-i\hbar\omega_\ell + \hbar\omega)e^{i\omega_\ell 0^+} & 0 \\ 0 & \zeta(i\hbar\omega_\ell + \hbar\omega)e^{-i\omega_\ell 0^+} \end{pmatrix} \begin{pmatrix} \alpha_\ell \\ \overline{\alpha}_{-\ell} \end{pmatrix} \quad \text{(4.30a)}$$

or, equivalently,

$$S = \beta\hbar \sum_\ell \overline{\alpha}_\ell(-i\hbar\omega_\ell + \hbar\omega)e^{i\omega_\ell 0^+}\alpha_\ell$$

$$= \frac{\beta\hbar}{2} \sum_\ell \left[\overline{\alpha}_\ell(-i\hbar\omega_\ell + \hbar\omega)e^{i\omega_\ell 0^+}\alpha_\ell + \alpha_{-\ell}\zeta(i\hbar\omega_\ell + \hbar\omega)e^{-i\omega_\ell 0^+}\overline{\alpha}_{-\ell}\right]$$

$$= \frac{\beta\hbar}{2} \sum_\ell \left(\overline{\alpha}_\ell \quad \alpha_{-\ell}\right) \begin{pmatrix} (-i\hbar\omega_\ell + \hbar\omega)e^{i\omega_\ell 0^+} & 0 \\ 0 & \zeta(i\hbar\omega_\ell + \hbar\omega)e^{-i\omega_\ell 0^+} \end{pmatrix} \begin{pmatrix} \alpha_\ell \\ \overline{\alpha}_{-\ell} \end{pmatrix}. \quad \text{(4.30b)}$$

Denoting the matrix in the last line as $-\mathcal{G}^{-1}(\ell)$, and taking into account that $\alpha_\ell$ and $\alpha_{-\ell}$ are not independent integration variables, the corresponding partition function is

$$\mathcal{Z} = \int \prod_\ell' \frac{d\overline{\alpha}_\ell d\alpha_\ell d\overline{\alpha}_{-\ell} d\alpha_{-\ell}}{(2\pi i)^{2\delta_{\zeta,1}}} \exp\left\{-\left(\overline{\alpha}_\ell \quad \alpha_{-\ell}\right)[-\beta\mathcal{G}^{-1}(\ell)]\begin{pmatrix} \alpha_\ell \\ \overline{\alpha}_{-\ell} \end{pmatrix}\right\}$$

$$= \prod_\ell' \det[-\beta\mathcal{G}^{-1}(\ell)]^{-\zeta}, \quad \text{(4.31)}$$

where the prime indicates that the integration is restricted to half of the frequency space, to prevent overcounting the fields. Therefore

$$-\zeta \ln \mathcal{Z} = \frac{1}{2} \sum_\ell \ln \det\left[-\beta\mathcal{G}^{-1}(\ell)\right]$$

$$= \frac{1}{2} \sum_\ell \left\{\ln\left[-\beta\mathcal{G}_{11}^{-1}(\ell)\right] + \ln\left[-\beta\mathcal{G}_{22}^{-1}(\ell)\right]\right\}$$

$$= \frac{1}{2} \sum_\ell \left\{\ln\left[\beta\hbar(-i\omega_\ell + \omega)e^{i\omega_\ell 0^+}\right] + \ln\left[\beta\hbar(i\omega_\ell + \omega)e^{-i\omega_\ell 0^+}\right] + \ln\zeta\right\}. \quad \text{(4.32)}$$

The last term is zero in the bosonic case ($\zeta = 1$), whereas in the fermionic case ($\zeta = -1$) it is $i\pi$. This is an unimportant pure number that can be neglected, or reabsorbed in the path integral measure. The final result is therefore

$$-\zeta \ln \mathcal{Z} = \sum_\ell \ln\left[\beta\hbar(-i\omega_\ell + \omega)e^{i\omega_\ell 0^+}\right], \quad \text{(4.33)}$$

which is again Eq. (4.4). The first line of Eq. (4.32) corresponds to the last line of Eq. (4.22). Thus we see that to render summations of this type well defined, we need to retrace the steps that led us to Eq. (4.22), writing the matrix determinant in terms of its component contributions, each with its own convergence factor according to the original time-ordering.

One may also choose to write the action in a matrix form such that both components appear with the same convergence factor [4]. In order to do so, let us consider the action

$$S^- = \int_0^{\beta\hbar} d\tau \, \alpha(\tau^+)\zeta(-\hbar\partial_\tau + \hbar\omega)\overline{\alpha}(\tau) = \beta\hbar \sum_\ell \alpha_{-\ell}\zeta(i\hbar\omega_\ell + \hbar\omega)e^{i\omega_\ell 0^+}\overline{\alpha}_{-\ell}. \quad \text{(4.34)}$$

We have [see Eq. (4.28)]

$$-\zeta \ln \mathcal{Z}^- = \lim_{\delta \to 0^+} \sum_\ell \ln[\beta\hbar(i\omega_\ell + \omega)]e^{i\omega_\ell\delta} = \ln\left(e^{\beta\hbar\omega} - \zeta\right), \qquad (4.35)$$

therefore the partition function $\mathcal{Z}^-$ is related to the partition function $\mathcal{Z} = (1 - \zeta e^{-\beta\hbar\omega})^{-\zeta}$ of the action $S$ in the first line of Eq. (4.30a) by $\mathcal{Z}^- = e^{-\zeta\beta\hbar\omega}\mathcal{Z}$. This gives us a relation between the actions themselves, $S = -\zeta\beta\hbar^2\omega + S^-$. We can now use this relation to write the action $S_{22}$ appearing as the 22 component of the matrix form in the second line of Eq. (4.30a) in terms of the corresponding $S_{22}^-$, obtaining

$$\begin{aligned}
S &= -\zeta\hbar\frac{\beta\hbar\omega}{2} + \frac{1}{2}\int_0^{\beta\hbar} d\tau \left(\overline{\alpha}(\tau^+) \quad \alpha(\tau^+)\right)\begin{pmatrix} \hbar\partial_\tau + \hbar\omega & 0 \\ 0 & \zeta(-\hbar\partial_\tau + \hbar\omega) \end{pmatrix}\begin{pmatrix} \alpha(\tau) \\ \overline{\alpha}(\tau) \end{pmatrix} \\
&= -\zeta\hbar\frac{\beta\hbar\omega}{2} + \frac{\beta\hbar}{2}\sum_\ell \left(\overline{\alpha}_\ell \quad \alpha_{-\ell}\right)\begin{pmatrix} (-i\hbar\omega_\ell + \hbar\omega)e^{i\omega_\ell 0^+} & 0 \\ 0 & \zeta(i\hbar\omega_\ell + \hbar\omega)e^{i\omega_\ell 0^+} \end{pmatrix}\begin{pmatrix} \alpha_\ell \\ \overline{\alpha}_{-\ell} \end{pmatrix}.
\end{aligned}$$
$$(4.36)$$

Comparing Eqs. (4.30a) and (4.36), we see that we have effectively changed the time-ordering of the fields of the 22 component at the cost of adding a constant term to the action, and both components now appear with the same convergence factor in frequency space. When performing the Gaussian integration, the contribution of the Gaussian part of the action to $-\zeta \ln \mathcal{Z}$ is then given by Eq. (4.29), while the contribution of the constant part of the action exactly cancels the spurious term $\beta\hbar\omega/2$, so that we obtain again the exact result

$$-\zeta \ln \mathcal{Z} = -\frac{\beta\hbar\omega}{2} + \frac{1}{2}\lim_{\delta \to 0^+}\sum_\ell \ln\left[\beta^2\hbar^2(\omega_\ell^2 + \omega^2)\right]e^{i\omega_\ell\delta} = \ln\left(1 - \zeta e^{-\beta\hbar\omega}\right). \qquad (4.37)$$

The preceding discussion should further clarify that the specific form of the convergence factors is determined by the time-ordering of the path integral. While the physical time-ordering is fixed by the construction of the discretized path integral, one may alter the time-ordering, and consequently the convergence factors, at the cost of introducing counterterms in the action. These counterterms ensure that the final result remains identical to that obtained using the original time-ordering. There are two reasons why one might prefer writing the action in the form of Eq. (4.36). The first, more formal, is that the two-component field $A(\tau) = \left(\alpha(\tau) \quad \overline{\alpha}(\tau)\right)^\mathsf{T}$ and its conjugate are then assigned a single, well-defined time argument: both components of $A$ are evaluated at the same imaginary time $\tau$, while both components of $\overline{A}$ are evaluated at the slightly later time $\tau^+$. Hence the two-component field can really be considered as a single object, and the Gaussian part of the action (4.36) takes the canonical form $\int_0^{\beta\hbar} d\tau \overline{A}(\tau^+)[-\mathcal{G}^{-1}(\tau)]A(\tau)$. The second, more practical, reason is that performing the Matsubara-frequency summations may turn out to be easier when both components carry the same convergence factor. We will see some examples of this in the following sections.

## 5 Weakly-interacting Bose gas

We consider now the $(D+1)$-dimensional quantum field theory for a nonrelativistic system of interacting bosons. The many-particle Hamiltonian is in general

$$\begin{aligned}
\hat{H} &= \int d^D\mathbf{x}\, \hat{\Psi}^\dagger(\mathbf{x})\left[-\frac{\hbar^2\nabla^2}{2m} + U(\mathbf{x})\right]\hat{\Psi}(\mathbf{x}) \\
&+ \frac{1}{2}\int d^D\mathbf{x}\, d^D\mathbf{x}'\, \hat{\Psi}^\dagger(\mathbf{x})\hat{\Psi}^\dagger(\mathbf{x}')V(\mathbf{x} - \mathbf{x}')\hat{\Psi}(\mathbf{x}')\hat{\Psi}(\mathbf{x}),
\end{aligned}$$
$$(5.1)$$

where $\hat{\Psi}(\mathbf{x})$, $\hat{\Psi}^\dagger(\mathbf{x})$ are bosonic field operators, $U(\mathbf{x})$ is an external potential, and $V(\mathbf{x} - \mathbf{x}')$ is the interaction potential. The number operator $\hat{N} = \int d^D\mathbf{x}\,\hat{\Psi}^\dagger(\mathbf{x})\hat{\Psi}(\mathbf{x})$ and the momentum operator $\hat{P} = -i\hbar \int d^D\mathbf{x}\,\hat{\Psi}^\dagger(\mathbf{x})\boldsymbol{\nabla}\hat{\Psi}(\mathbf{x})$ commute with the Hamiltonian and are therefore conserved in all physical processes. Our goal will be to compute the grand canonical partition function

$$\mathcal{Z} = \mathrm{Tr}\left[e^{-\beta(\hat{H}-\mu\hat{N})}\right] \equiv \mathrm{Tr}(e^{-\beta\mathscr{H}}). \tag{5.2}$$

The system can be considered weakly interacting if the diluteness condition $|a_s|^D N/L^D \ll 1$ holds, where $a_s$ is the $s$-wave scattering length and $L$ is the linear size of the system. In this scenario, a perturbative treatment of $\hat{H}$ is viable. At small temperatures, the standard perturbative approach introduced by Bogoliubov [1, 19–24] is based on the assumption that the system exhibits Bose-Einstein condensation (BEC), namely a macroscopic occupation of a single one-particle state described by the normalized wavefunction $\chi_0(\mathbf{x})$. Writing the field operator in terms of annihilation operators as

$$\hat{\Psi}(\mathbf{x}) \equiv \hat{\Psi}_0(\mathbf{x}) + \hat{\eta}(\mathbf{x}) = \chi_0(\mathbf{x})\hat{a}_0 + \sum_{i\neq 0}\chi_i(\mathbf{x})\hat{a}_i, \tag{5.3}$$

the macroscopic occupation of $\chi_0(\mathbf{x})$ is realized by means of the Bogoliubov prescription $\hat{a}_0 \to \sqrt{N_0}$, where $N_0$ is the occupation number of $\chi_0(\mathbf{x})$. This amounts to replace the zero-mode component of the field operator by the classical field

$$\Psi_0(\mathbf{x}) = \sqrt{N_0}\chi_0(\mathbf{x}), \tag{5.4}$$

which defines the BEC order parameter. At the mean-field level, this is related to the chemical potential $\mu$ of the system by

$$\mu = \frac{1}{N_0}\int d^D\mathbf{x}\,\Psi_0^*(\mathbf{x})\left[-\frac{\hbar^2\nabla^2}{2m} + U(\mathbf{x}) + \int d^D\mathbf{x}'\,V(\mathbf{x}-\mathbf{x}')|\Psi_0(\mathbf{x}',t)|^2\right]\Psi_0(\mathbf{x}), \tag{5.5}$$

and its time evolution is given by the Gross-Pitaevskii equation [25, 26]

$$i\hbar\frac{\partial}{\partial t}\Psi_0(\mathbf{x},t) = \left[-\frac{\hbar^2\nabla^2}{2m} + U(\mathbf{x}) + \int d^D\mathbf{x}'\,V(\mathbf{x}-\mathbf{x}')|\Psi_0(\mathbf{x}',t)|^2\right]\Psi_0(\mathbf{x},t). \tag{5.6}$$

For simplicity, in the following we specialize the Hamiltonian (5.1) to the case of a uniform system by setting $U(\mathbf{x}) = 0$, while allowing for an arbitrary repulsive interaction potential

$$V(\mathbf{x}-\mathbf{x}') = \frac{1}{L^D}\sum_{\mathbf{k}}e^{i\mathbf{k}\cdot(\mathbf{x}-\mathbf{x}')}\widetilde{V}(\mathbf{k}), \tag{5.7}$$

where $\widetilde{V}(\mathbf{k}) = \int d^D\mathbf{x}\,e^{-i\mathbf{k}\cdot\mathbf{x}}V(\mathbf{x}) \geq 0$ is the corresponding Fourier transform.

## 5.1 Hamiltonian approach

The field operator can be expanded on the basis of eigenfunctions $\{\psi_\mathbf{k}(\mathbf{x})\}$ of the one-particle Hamiltonian $\hat{h} = -\hbar^2\nabla^2/2m$, which are plane waves indexed by the wave vector $\mathbf{k}$, as

$$\hat{\Psi}(\mathbf{x}) = \sum_{\mathbf{k}}\psi_\mathbf{k}(\mathbf{x})\hat{a}_\mathbf{k} = \sum_{\mathbf{k}}\frac{e^{i\mathbf{k}\cdot\mathbf{x}}}{\sqrt{L^D}}\hat{a}_\mathbf{k}. \tag{5.8}$$

This corresponds to a change of basis in the one-particle Hilbert space, from the position basis to the momentum basis. The Hamiltonian then becomes

$$\hat{H} = \sum_{\mathbf{k}}\epsilon_k\hat{a}_\mathbf{k}^\dagger\hat{a}_\mathbf{k} + \frac{1}{2L^D}\sum_{\mathbf{k}\mathbf{k}'\mathbf{q}}\widetilde{V}(\mathbf{q})\hat{a}_{\mathbf{k}+\mathbf{q}}^\dagger\hat{a}_{\mathbf{k}'-\mathbf{q}}^\dagger\hat{a}_{\mathbf{k}'}\hat{a}_\mathbf{k}, \tag{5.9}$$

where $k = |\mathbf{k}|$ and

$$\epsilon_k = \frac{\hbar^2 k^2}{2m} \tag{5.10}$$

is the one-particle energy.

A rigorous result is that for a uniform noninteracting Bose gas below the critical temperature for BEC, the macroscopically occupied one-particle state is the ground state of the one-particle Hamiltonian, i.e. the state with $\mathbf{k} = \mathbf{0}$. We assume the same for the uniform weakly-interacting gas, so that the Bogoliubov prescription is

$$\hat{\Psi}(\mathbf{x}) = \Psi_0 + \sum_{\mathbf{k} \neq \mathbf{0}} \frac{e^{i\mathbf{k} \cdot \mathbf{x}}}{\sqrt{L^D}} \hat{a}_{\mathbf{k}}, \tag{5.11}$$

where

$$\Psi_0 = \sqrt{\frac{N_0}{L^D}} \equiv \sqrt{n_0}. \tag{5.12}$$

Correspondingly, the number operator is

$$\hat{N} = N_0 + \sum_{\mathbf{k} \neq \mathbf{0}} \hat{N}_{\mathbf{k}} = N_0 + \sum_{\mathbf{k} \neq \mathbf{0}} \hat{a}_{\mathbf{k}}^\dagger \hat{a}_{\mathbf{k}}. \tag{5.13}$$

Substituting $\hat{a}_{\mathbf{0}} \to \sqrt{N_0}$ in Eq. (5.9) and retaining only the interaction terms that are no more than quadratic in $\hat{a}_{\mathbf{k}}$ and $\hat{a}_{\mathbf{k}}^\dagger$ operators with $\mathbf{k} \neq \mathbf{0}$ (Bogoliubov approximation, also called Gaussian or one-loop), we obtain $\hat{H} \simeq \hat{H}_G$, where

$$\hat{H}_G = E_0 + \sum_{\mathbf{k} \neq \mathbf{0}} \left\{ \left[ \epsilon_k + n_0 \widetilde{V}(\mathbf{0}) + \frac{n_0}{2} \left( \widetilde{V}(\mathbf{k}) + \widetilde{V}(-\mathbf{k}) \right) \right] \hat{a}_{\mathbf{k}}^\dagger \hat{a}_{\mathbf{k}} + \frac{n_0 \widetilde{V}(\mathbf{k})}{2} (\hat{a}_{\mathbf{k}}^\dagger \hat{a}_{-\mathbf{k}}^\dagger + \hat{a}_{\mathbf{k}} \hat{a}_{-\mathbf{k}}) \right\} \tag{5.14}$$

and

$$E_0 = L^D \frac{\widetilde{V}(\mathbf{0}) n_0^2}{2} \tag{5.15}$$

is the mean-field ground state energy. It follows that $\hat{\mathscr{H}}_G \equiv \hat{H}_G - \mu \hat{N}$ is

$$\hat{\mathscr{H}}_G = \Omega_0 + \sum_{\mathbf{k} \neq \mathbf{0}}' \left\{ \left[ \epsilon_k + n_0 \widetilde{V}(\mathbf{0}) - \mu + \frac{n_0}{2} \left( \widetilde{V}(\mathbf{k}) + \widetilde{V}(-\mathbf{k}) \right) \right] (\hat{a}_{\mathbf{k}}^\dagger \hat{a}_{\mathbf{k}} + \hat{a}_{-\mathbf{k}}^\dagger \hat{a}_{-\mathbf{k}}) \right.$$
$$\left. + n_0 \widetilde{V}(\mathbf{k}) (\hat{a}_{\mathbf{k}}^\dagger \hat{a}_{-\mathbf{k}}^\dagger + \hat{a}_{\mathbf{k}} \hat{a}_{-\mathbf{k}}) \right\}, \tag{5.16}$$

where

$$\Omega_0 = E_0 - \mu N_0 \tag{5.17}$$

and the prime indicates that the summation is restricted to one half of momentum space, since the terms corresponding to $\mathbf{k}$ and $-\mathbf{k}$ must be counted only once[5].

The Hamiltonian (5.16) can be diagonalized via the Bogoliubov transformation, that is the pseudorotation [20, 23]

$$\begin{pmatrix} \hat{b}_{\mathbf{k}} \\ \hat{b}_{-\mathbf{k}}^\dagger \end{pmatrix} = \begin{pmatrix} u_{\mathbf{k}} & -v_{\mathbf{k}} \\ -v_{\mathbf{k}} & u_{\mathbf{k}} \end{pmatrix} \begin{pmatrix} \hat{a}_{\mathbf{k}} \\ \hat{a}_{-\mathbf{k}}^\dagger \end{pmatrix}, \qquad \begin{pmatrix} \hat{a}_{\mathbf{k}} \\ \hat{a}_{-\mathbf{k}}^\dagger \end{pmatrix} = \begin{pmatrix} u_{\mathbf{k}} & v_{\mathbf{k}} \\ v_{\mathbf{k}} & u_{\mathbf{k}} \end{pmatrix} \begin{pmatrix} \hat{b}_{\mathbf{k}} \\ \hat{b}_{-\mathbf{k}}^\dagger \end{pmatrix}, \tag{5.18}$$

---

[5]We notice that, at the level of Gaussian approximation, it does not make any difference whether the condensate density $n_0$ or the total density $n$ appears in summation, since one could use Eq. (5.13) to write $N_0$ in terms of $N$ up to corrections of quartic order in the $\hat{a}_{\mathbf{k}}$ and $\hat{a}_{\mathbf{k}}^\dagger$ operators. However, it has been suggested on variational grounds that the second possibility provides a more accurate result for the grand potential and the condensate fraction [27, 28]. Nevertheless, here we will stick to the standard treatment, where there is $n_0$ and not $n$.

where $\hat{b}_\mathbf{k}$, $\hat{b}_\mathbf{k}^\dagger$ are new bosonic operators satisfying canonical commutation relations, provided that the real functions $u_\mathbf{k}$, $v_\mathbf{k}$ satisfy

$$u_\mathbf{k}^2 = v_\mathbf{k}^2 + 1 = \frac{1}{2}\left[\frac{\epsilon_k + n_0 \widetilde{V}(\mathbf{0}) - \mu + \frac{n_0}{2}\left(\widetilde{V}(\mathbf{k}) + \widetilde{V}(-\mathbf{k})\right)}{\mathcal{E}_\mathbf{k}(\mu, n_0)} + 1\right], \tag{5.19}$$

where

$$\mathcal{E}_\mathbf{k}(\mu, n_0) = \sqrt{\left[\epsilon_k + n_0 \widetilde{V}(\mathbf{0}) - \mu + \frac{n_0}{2}\left(\widetilde{V}(\mathbf{k}) + \widetilde{V}(-\mathbf{k})\right)\right]^2 - \left[n_0 \widetilde{V}(\mathbf{k})\right]^2} \tag{5.20}$$

is the generalized Bogoliubov spectrum. The diagonalized Hamiltonian reads

$$\mathscr{H}_G = \Omega_0 + \Omega_G^{(0)} + \sum_{\mathbf{k} \neq \mathbf{0}} \mathcal{E}_\mathbf{k} \hat{b}_\mathbf{k}^\dagger \hat{b}_\mathbf{k}, \tag{5.21}$$

where

$$\Omega_G^{(0)} = \frac{1}{2}\sum_{\mathbf{k} \neq \mathbf{0}}\left[\mathcal{E}_\mathbf{k} - \epsilon_k - n_0 \widetilde{V}(\mathbf{0}) + \mu - \frac{n_0}{2}\left(\widetilde{V}(\mathbf{k}) + \widetilde{V}(-\mathbf{k})\right)\right]. \tag{5.22}$$

This shows that, within the assumptions made, the original system of interacting bosons can be described by a Hamiltonian of noninteracting bosonic quasiparticles with the Bogoliubov spectrum $\mathcal{E}_\mathbf{k}$. The operators $\hat{b}_\mathbf{k}^\dagger$ and $\hat{b}_\mathbf{k}$ represent the creation and annihilation operators of these quasiparticles. From this perspective, a physical particle created by $\hat{a}_\mathbf{k}^\dagger$ is described as a superposition of quasiparticles, according to Eq. (5.18). At small momenta, $u_\mathbf{k} \sim \sqrt{mc_s/2\hbar k} \gg 1$, where $c_s = \sqrt{n_0 \widetilde{V}(\mathbf{0})/m}$ is the Bogoliubov sound velocity, and $v_\mathbf{k} \sim -u_\mathbf{k}$, therefore $\hat{a}_\mathbf{k}^\dagger \sim \sqrt{mc_s/2\hbar k}(\hat{b}_\mathbf{k}^\dagger - \hat{b}_{-\mathbf{k}})$ and a physical particle is described by a very large number of quasiparticles. This is equivalent to say that a single quasiparticle excitation corresponds to a collective excitation of many physical particles. Instead, at large momenta $u_\mathbf{k} \sim 1$ and $v_\mathbf{k} \sim 0$, so that $\hat{a}_\mathbf{k}^\dagger \sim \hat{b}_\mathbf{k}^\dagger$ and the quasiparticles become indistinguishable from the real particles.

In terms of the new quasiparticle operators, Eq. (5.13) becomes

$$\hat{N} = N_0 + \sum_{\mathbf{k} \neq \mathbf{0}} v_\mathbf{k}^2 + \sum_{\mathbf{k} \neq \mathbf{0}}(u_\mathbf{k}^2 + v_\mathbf{k}^2)\hat{b}_\mathbf{k}^\dagger \hat{b}_\mathbf{k} + \sum_{\mathbf{k} \neq \mathbf{0}} u_\mathbf{k} v_\mathbf{k}(\hat{b}_\mathbf{k}^\dagger \hat{b}_{-\mathbf{k}}^\dagger + \hat{b}_\mathbf{k} \hat{b}_{-\mathbf{k}}). \tag{5.23}$$

The expectation value of last term on the eigenstates of the Hamiltonian (5.21) vanishes, because the Hamiltonian commutes with each $\hat{N}_\mathbf{k}^{ex} \equiv \hat{b}_\mathbf{k}^\dagger \hat{b}_\mathbf{k}$. Therefore the number operator may be written equivalently without such term. Since the chemical potential is related to $n_0$ by [see Eqs. (5.5) and (5.12)]

$$\mu = n_0 \widetilde{V}(\mathbf{0}), \tag{5.24}$$

we then have

$$\hat{N} = N_0 + \frac{1}{2}\sum_{\mathbf{k} \neq \mathbf{0}}\frac{\epsilon_k + \frac{n_0}{2}\left(\widetilde{V}(\mathbf{k}) + \widetilde{V}(-\mathbf{k})\right) - \mathcal{E}_\mathbf{k}(n_0)}{\mathcal{E}_\mathbf{k}(n_0)} + \sum_{\mathbf{k} \neq \mathbf{0}}\frac{\epsilon_k + \frac{n_0}{2}\left(\widetilde{V}(\mathbf{k}) + \widetilde{V}(-\mathbf{k})\right)}{\mathcal{E}_\mathbf{k}(n_0)}\hat{b}_\mathbf{k}^\dagger \hat{b}_\mathbf{k}, \tag{5.25}$$

with

$$\mathcal{E}_\mathbf{k}(n_0) = \sqrt{\left[\epsilon_k + \frac{n_0}{2}\left(\widetilde{V}(\mathbf{k}) + \widetilde{V}(-\mathbf{k})\right)\right]^2 - \left[n_0 \widetilde{V}(\mathbf{k})\right]^2}. \tag{5.26}$$

Quasiparticles are noninteracting, thus the thermal average of their number $\langle N_\mathbf{k}^{ex} \rangle$ follows the Bose-Einstein distribution. This yields

$$N(N_0, \beta) = N_0 + \frac{1}{2}\sum_{\mathbf{k} \neq \mathbf{0}}\frac{\epsilon_k + \frac{n_0}{2}\left(\widetilde{V}(\mathbf{k}) + \widetilde{V}(-\mathbf{k})\right) - \mathcal{E}_\mathbf{k}(n_0)}{\mathcal{E}_\mathbf{k}(n_0)} + \sum_{\mathbf{k} \neq \mathbf{0}}\frac{\epsilon_k + \frac{n_0}{2}\left(\widetilde{V}(\mathbf{k}) + \widetilde{V}(-\mathbf{k})\right)}{(e^{\beta \mathcal{E}_\mathbf{k}(n_0)} - 1)\mathcal{E}_\mathbf{k}(n_0)}, \tag{5.27}$$

which is an implicit equation for the condensate fraction $N_0/N$ at finite temperature [28].

At this point, the grand canonical partition function is computed from Eq. (5.21) as

$$
\begin{aligned}
\mathcal{Z}_G &= \text{Tr}\left(e^{-\beta\hat{\mathcal{H}}_G}\right) \\
&= e^{-\beta(\Omega_0+\Omega_G^{(0)})} \sum_{\{N_{\mathbf{k}}^{ex}\}} \langle\{N_{\mathbf{k}}^{ex}\}|e^{-\beta\sum_{\mathbf{k}\neq 0}\mathcal{E}_{\mathbf{k}}\hat{N}_{\mathbf{k}}^{ex}}|\{N_{\mathbf{k}}^{ex}\}\rangle \\
&= e^{-\beta(\Omega_0+\Omega_G^{(0)})} \prod_{\mathbf{k}\neq\mathbf{0}} \frac{1}{1-e^{-\beta\mathcal{E}_{\mathbf{k}}}},
\end{aligned}
\tag{5.28}
$$

which gives the grand potential

$$
\begin{aligned}
\Omega_G &\equiv -\frac{1}{\beta}\ln\mathcal{Z}_G \\
&= \Omega_0 + \frac{1}{2}\sum_{\mathbf{k}\neq\mathbf{0}}\left[\mathcal{E}_{\mathbf{k}}-\epsilon_k-n_0\widetilde{V}(\mathbf{0})+\mu-\frac{n_0}{2}\left(\widetilde{V}(\mathbf{k})+\widetilde{V}(-\mathbf{k})\right)\right] + \frac{1}{\beta}\sum_{\mathbf{k}\neq\mathbf{0}}\ln\left(1-e^{-\beta\mathcal{E}_{\mathbf{k}}}\right).
\end{aligned}
\tag{5.29}
$$

One can easily verify that $N = -\partial\Omega_G/\partial\mu$ evaluated in $\mu = n_0\widetilde{V}(\mathbf{0})$ gives back Eq. (5.27).

*Zero-range interaction*—In the case of a repulsive zero-range interaction modeled by a delta function potential $V(\mathbf{x}-\mathbf{x}') = g\delta^D(\mathbf{x}-\mathbf{x}')$, with $g > 0$, we have $\widetilde{V}(\mathbf{k}) = g = \text{const.}$, hence the above result simplifies to

$$
\Omega_G = \Omega_0 + \frac{1}{2}\sum_{\mathbf{k}\neq\mathbf{0}}(\mathcal{E}_k-\epsilon_k-2gn_0+\mu) + \frac{1}{\beta}\sum_{\mathbf{k}\neq\mathbf{0}}\ln\left(1-e^{-\beta\mathcal{E}_k}\right),
\tag{5.30}
$$

with

$$
\mathcal{E}_k(\mu,n_0) = \sqrt{(\epsilon_k+2gn_0-\mu)^2-(gn_0)^2}.
\tag{5.31}
$$

## 5.2 Path integral approach

Now we will show how exactly the same result can be obtained with the path integral approach. The grand canonical partition function is [Eq. (2.20)]

$$
\mathcal{Z} = \int \mathcal{D}\Psi^*\mathcal{D}\Psi\, e^{-S[\Psi,\Psi^*]/\hbar} = \int \mathcal{D}\Psi^*\mathcal{D}\Psi\, e^{-\frac{1}{\hbar}\int_0^{\beta\hbar}d\tau\int d^D\mathbf{x}\mathcal{L}[\Psi^*,\Psi]},
\tag{5.32}
$$

where

$$
\mathcal{L} = \Psi^*(\mathbf{x},\tau)\left(\hbar\partial_\tau-\frac{\hbar^2\nabla^2}{2m}-\mu\right)\Psi(\mathbf{x},\tau) + \frac{1}{2}\int d^D\mathbf{x}'\, V(\mathbf{x}-\mathbf{x}')|\Psi(\mathbf{x},\tau)|^2|\Psi(\mathbf{x}',\tau)|^2
\tag{5.33}
$$

is the Lagrangian density of the system, and $\Psi(\mathbf{x})$ are the eigenfunctions associated to the bosonic coherent states $|\Psi\rangle \propto e^{\int d^D\mathbf{x}\Psi(\mathbf{x})\hat{\Psi}^\dagger(\mathbf{x})}|0\rangle$ that we use as the representation basis for the path integral [cf. Eq. (2.18)]. Following the Bogoliubov prescription (5.11), the eigenfunctions $\Psi(\mathbf{x},\tau)$ are separated as[6]

$$
\Psi(\mathbf{x},\tau) = \Psi_0 + \eta(\mathbf{x},\tau).
\tag{5.34}
$$

The part of the Lagrangian that depends only on $\Psi_0$ and $\Psi_0^*$ reads

$$
\mathcal{L}_0 = -\mu|\Psi_0|^2 + \frac{\widetilde{V}(\mathbf{0})}{2}|\Psi_0|^4, \qquad \widetilde{V}(\mathbf{0}) > 0.
\tag{5.35}
$$

---

[6]The reason for $\Psi_0$ to be time independent is that in the construction of the path integral, time-slicing and the introduction of time-dependent integration variables were required because the Hamiltonian contained non-commuting operators. Since the Bogoliubov prescription replaces the quantum operator $\hat{\Psi}_0$ with a classical quantity, the corresponding integration variable in the path integral is time independent.

For large $|\Psi_0|$, the quartic term dominates and guarantees the action $S_0 = \beta \hbar L^D \mathcal{L}_0$ to be bounded from below, and thus the stability of the system, for any value of $\mu$. In particular, for $\mu \leq 0$, $S_0$ has a global minimum at $|\Psi_0| = 0$, which means that there is no stable condensate amplitude. Instead, for $\mu > 0$, $S_0$ has the shape of a "Mexican hat", with a full circle of degenerate minima at $|\Psi_0|^2 = \mu/\widetilde{V}(\mathbf{0})$, corresponding to a finite condensate density $n_0 = \mu/\widetilde{V}(\mathbf{0})$, in agreement with Eq. (5.5). This condition fixes $\Psi_0 = \sqrt{\mu/\widetilde{V}(\mathbf{0})}\, e^{i\theta}$, with $\theta \in S^1$. By fixing the phase $\theta$ we select a particular minimum, determining the spontaneous breakdown of the $U(1)$ symmetry of the action. Without loss of generality, we take $\theta = 0$, so that $\Psi_0$ is real and positive, corresponding to $\Psi_0 = \sqrt{n_0} = \sqrt{\mu/\widetilde{V}(\mathbf{0})}$.

Analogously to what we did in the Hamiltonian approach, we now substitute Eq. (5.34) into Eq. (5.33) and retain only terms up to second order in the fluctuations $\eta$, $\eta^*$ around $\Psi_0$ (Gaussian or one-loop approximation). The condition $\Psi_0 = \sqrt{\mu/\widetilde{V}(\mathbf{0})}$ ensures that we are expanding around a minimum of the action, hence linear terms in the fluctuations vanish. Expanding the fluctuations in Fourier series as

$$\eta(\mathbf{x}, \tau) = \frac{1}{\sqrt{L^D}} \sum_{\mathbf{k} \neq \mathbf{0}} \sum_{n=-\infty}^{\infty} a_{\mathbf{k},n}\, e^{i(\mathbf{k}\cdot\mathbf{x}-\omega_n\tau)}, \qquad \omega_n = \frac{2\pi n}{\beta\hbar}, \tag{5.36}$$

and taking into account the time-ordering, we obtain

$$S \simeq S_G = \beta\hbar L^D \mathcal{L}_0 + \frac{\beta\hbar}{2} \sum_q \begin{pmatrix} a_q^* & a_{-q} \end{pmatrix} \begin{pmatrix} -\mathcal{G}_{11}^{-1}(q)e^{i\omega_n 0^+} & -\mathcal{G}_{12}^{-1}(q) \\ -\mathcal{G}_{21}^{-1}(q) & -\mathcal{G}_{22}^{-1}(q)e^{-i\omega_n 0^+} \end{pmatrix} \begin{pmatrix} a_q \\ a_{-q}^* \end{pmatrix}, \tag{5.37}$$

where

$$-\mathcal{G}_{11}^{-1}(q) = -i\hbar\omega_n + \epsilon_k + n_0 \widetilde{V}(\mathbf{0}) - \mu + \frac{n_0}{2}\left(\widetilde{V}(\mathbf{k}) + \widetilde{V}(-\mathbf{k})\right), \tag{5.38a}$$

$$-\mathcal{G}_{12}^{-1}(q) = -\mathcal{G}_{21}^{-1}(q) = n_0 \widetilde{V}(\mathbf{k}), \tag{5.38b}$$

$$-\mathcal{G}_{22}^{-1}(q) = i\hbar\omega_n + \epsilon_k + n_0 \widetilde{V}(\mathbf{0}) - \mu + \frac{n_0}{2}\left(\widetilde{V}(\mathbf{k}) + \widetilde{V}(-\mathbf{k})\right), \tag{5.38c}$$

and $q = (\mathbf{k}, n)$ with $\mathbf{k} \neq \mathbf{0}$. $\mathcal{G}(q)$ is the classical propagator of the $\eta$ field in Fourier space. Its poles, that are determined by the condition $\det[-\mathcal{G}^{-1}(q)] = 0$, yield the dispersion relation of the excitations, which is again the generalized Bogoliubov spectrum (5.20).

To compute the partition function we must now perform the path integral over the fluctuation fields, following the procedure illustrated in Section 4.2. The measure is

$$\int \mathcal{D}\eta^* \mathcal{D}\eta = \int \prod_{(\mathbf{x},\tau)} \frac{d\eta^*(\mathbf{x},\tau)d\eta(\mathbf{x},\tau)}{2\pi i/L^D} = \int \prod_q' \frac{da_q^* da_q da_{-q}^* da_{-q}}{(2\pi i)^2}, \tag{5.39}$$

where the prime indicates that the product is restricted to one half of $\mathbf{k}$ space. Hence

$$\mathcal{Z}_G = e^{-\beta L^D \mathcal{L}_0} \int \prod_q' \frac{da_q^* da_q da_{-q}^* da_{-q}}{(2\pi i)^2} \exp\left\{ -\sum_q' \begin{pmatrix} a_q^* & a_{-q} \end{pmatrix} \left[-\beta\mathcal{G}^{-1}(q)\right] \begin{pmatrix} a_q \\ a_{-q}^* \end{pmatrix} \right\}$$

$$= e^{-\beta L^D \mathcal{L}_0} \prod_q' \det\left[-\beta\mathcal{G}^{-1}(q)\right]^{-1}. \tag{5.40}$$

Here we have used that fact that the matrix $-\mathcal{G}^{-1}(q)$, although it is not Hermitian, has a positive-definite Hermitian part. Furthermore, its determinant is manifestly real, ensuring

that the partition function is real. We then obtain

$$
\Omega_G = \Omega_0 + \frac{1}{\beta} {\sum_q}' \ln \det \left[ -\beta \mathcal{G}^{-1}(q) \right]
$$

$$
= \Omega_0 + \frac{1}{2\beta} \sum_q \ln \left[ \beta^2 \mathcal{G}_{11}^{-1}(q) \mathcal{G}_{22}^{-1}(q) - \beta^2 \mathcal{G}_{21}^{-1}(q) \mathcal{G}_{12}^{-1}(q) \right]
$$

$$
= \Omega_0 + \frac{1}{2\beta} \sum_q \left\{ \ln \left[ -\beta \mathcal{G}_{11}^{-1}(q) e^{i\omega_n 0^+} \right] + \ln \left[ -\beta \mathcal{G}_{22}^{-1}(q) e^{-i\omega_n 0^+} \right] + \ln \left[ 1 - \frac{\mathcal{G}_{21}^{-1}(q) \mathcal{G}_{12}^{-1}(q)}{\mathcal{G}_{11}^{-1}(q) \mathcal{G}_{22}^{-1}(q)} \right] \right\},
$$

$$
\tag{5.41}
$$

where $\Omega_0 = L^D \mathcal{L}_0 = L^D \widetilde{V}(\mathbf{0}) n_0^2 / 2 - \mu N_0$, in agreement with Eq. (5.17). The last term in the summation is convergent, but it is convenient to multiply it by $e^{i\omega_n \delta}$, which does not change the result [6]. Exploiting the fact that $\mathcal{G}_{22}^{-1}(q) = \mathcal{G}_{11}^{-1}(-q)$, the summation is then equal to

$$
\lim_{\delta \to 0^+} \sum_q \left\{ 2 \ln \left[ -\beta \mathcal{G}_{11}^{-1}(q) \right] + \ln \left[ 1 - \frac{\mathcal{G}_{21}^{-1}(q) \mathcal{G}_{12}^{-1}(q)}{\mathcal{G}_{11}^{-1}(q) \mathcal{G}_{22}^{-1}(q)} \right] \right\} e^{i\omega_n \delta} =
$$

$$
\lim_{\delta \to 0^+} \sum_q \left\{ \ln \left[ -\beta \mathcal{G}_{11}^{-1}(q) \right] - \ln \left[ -\beta \mathcal{G}_{22}^{-1}(q) \right] + \ln \det \left[ -\beta \mathcal{G}^{-1}(q) \right] \right\} e^{i\omega_n \delta}.
\tag{5.42}
$$

Using [see Eqs. (4.7), (4.35), and (4.29), respectively]

$$
\lim_{\delta \to 0^+} \sum_q \ln \left[ -\beta \mathcal{G}_{11}^{-1}(q) \right] e^{i\omega_n \delta} = \sum_{\mathbf{k} \neq \mathbf{0}} \ln \left\{ 1 - e^{-\beta \left[ \epsilon_k + n_0 \widetilde{V}(\mathbf{0}) - \mu + \frac{n_0}{2} \left( \widetilde{V}(\mathbf{k}) + \widetilde{V}(-\mathbf{k}) \right) \right]} \right\},
\tag{5.43a}
$$

$$
\lim_{\delta \to 0^+} \sum_q \ln \left[ -\beta \mathcal{G}_{22}^{-1}(q) \right] e^{i\omega_n \delta} = \sum_{\mathbf{k} \neq \mathbf{0}} \ln \left\{ e^{\beta \left[ \epsilon_k + n_0 \widetilde{V}(\mathbf{0}) - \mu + \frac{n_0}{2} \left( \widetilde{V}(\mathbf{k}) + \widetilde{V}(-\mathbf{k}) \right) \right]} - 1 \right\},
\tag{5.43b}
$$

$$
\lim_{\delta \to 0^+} \sum_q \ln \det \left[ -\beta \mathcal{G}^{-1}(q) \right] e^{i\omega_n \delta} = \sum_{\mathbf{k} \neq \mathbf{0}} \left[ \beta \mathcal{E}_{\mathbf{k}} + 2 \ln \left( 1 - e^{-\beta \mathcal{E}_{\mathbf{k}}} \right) \right],
\tag{5.43c}
$$

we finally obtain

$$
\Omega_G = \Omega_0 + \frac{1}{2} \sum_{\mathbf{k} \neq \mathbf{0}} \left[ \mathcal{E}_{\mathbf{k}} - \epsilon_k - n_0 \widetilde{V}(\mathbf{0}) + \mu - \frac{n_0}{2} \left( \widetilde{V}(\mathbf{k}) + \widetilde{V}(-\mathbf{k}) \right) \right] + \frac{1}{\beta} \sum_{\mathbf{k} \neq \mathbf{0}} \ln(1 - e^{-\beta \mathcal{E}_{\mathbf{k}}}),
\tag{5.44}
$$

which coincides with the Hamiltonian result, Eq. (5.29). We note that an undoubted advantage of the path integral over the Hamiltonian approach is the possibility of avoiding the complications related to the Bogoliubov transformation.

*Alternative time-ordering*—Alternatively, we can change the time-ordering so that both components of the matrix in Eq. (5.37) appear with the same convergence factor. As described in Section 4.2, the action then becomes

$$
S_G = \beta \hbar L^D \mathcal{L}_0 - \frac{\beta \hbar}{2} \sum_{\mathbf{k} \neq \mathbf{0}} \left[ \epsilon_k + n_0 \widetilde{V}(\mathbf{0}) - \mu + \frac{n_0}{2} \left( \widetilde{V}(\mathbf{k}) + \widetilde{V}(-\mathbf{k}) \right) \right]
$$

$$
+ \frac{\beta \hbar}{2} \sum_q \begin{pmatrix} a_q^* & a_{-q} \end{pmatrix} \begin{pmatrix} -\mathcal{G}_{11}^{-1}(q) e^{i\omega_n 0^+} & -\mathcal{G}_{12}^{-1}(q) \\ -\mathcal{G}_{21}^{-1}(q) & -\mathcal{G}_{22}^{-1}(q) e^{i\omega_n 0^+} \end{pmatrix} \begin{pmatrix} a_q \\ a_{-q}^* \end{pmatrix}.
\tag{5.45}
$$

The corresponding grand potential is

$$
\Omega = \Omega_0 - \frac{1}{2} \sum_{\mathbf{k} \neq \mathbf{0}} \left[ \epsilon_k + n_0 \widetilde{V}(\mathbf{0}) - \mu + \frac{n_0}{2} \left( \widetilde{V}(\mathbf{k}) + \widetilde{V}(-\mathbf{k}) \right) \right] + \frac{1}{2\beta} \sum_q \ln \det \left[ -\beta \mathcal{G}^{-1}(q) \right] e^{i\omega_n 0^+},
\tag{5.46}
$$

and using result (5.43c) for the last summation, we immediately obtain Eq. (5.44). This alternative time-ordering allows us to obtain the result by calculating only one frequency summation instead of three.

The treatments we have presented, which are based on the careful treatment of convergence factors following the implicit time-ordering of the path integral, and which reproduce exactly the Hamiltonian result, are not the most common in the literature, even for the simpler case of a zero-range interaction. The usual route is to write Eq. (5.41) as

$$\Omega_G = \Omega_0 + \frac{1}{2\beta} \sum_q \ln\left[\beta^2 \left(\hbar^2 \omega_n^2 + \mathcal{E}_{\mathbf{k}}^2\right)\right]. \tag{5.47}$$

We considered summations of this form in Section 4.2 [see in particular Eq. (4.29)] and showed that a naive evaluation of the Matsubara summation yields the commonly quoted expression [18, 22, 27–30]

$$\Omega_G = \Omega_0 + \frac{1}{2} \sum_{\mathbf{k}\neq\mathbf{0}} \mathcal{E}_{\mathbf{k}} + \frac{1}{\beta} \sum_{\mathbf{k}\neq\mathbf{0}} \ln\left(1 - e^{-\beta\mathcal{E}_{\mathbf{k}}}\right). \tag{5.48}$$

This does not coincide with the result obtained from the Hamiltonian approach because certain contributions to $\Omega_G^{(0)}$ are missing. This discrepancy is not catastrophic in practice, since the zero-point energy must be regularized in any case [30–36], and after a suitable regularization Eqs. (5.29) and (5.48) lead to the same grand potential [30]. Nevertheless, the difference is conceptually important: when Matsubara summations are performed naively, the regularization procedures required for the Hamiltonian and path-integral calculations differ, and only the final, regularized results coincide. By contrast, the method advocated here yields agreement already at the intermediate stage, so that the subsequent regularization is identical in both approaches.

# 6  BCS Superconductor

We consider as a final example the Bardeen-Cooper-Schrieffer (BCS) theory of superconductivity [37–39]. The conceptual foundation of the theory is the existence of an effective attractive interaction between electrons mediated by vibrations of the crystal lattice (phonons), which induces an instability of the electron gas towards the formation of bound states of two electrons (Cooper pairs) in the vicinity of the Fermi surface. Cooper pairs behave like composite bosonic quasiparticles, and at low temperatures they form a condensate which is responsible for conventional superconductivity.

## 6.1  Hamiltonian approach

The Hamiltonian of the system in momentum space is [40, 41]

$$\hat{H} = \sum_{\mathbf{k},\sigma} \epsilon_{\mathbf{k}} \hat{c}_{\mathbf{k},\sigma}^\dagger \hat{c}_{\mathbf{k},\sigma} + \frac{1}{2L^D} \sum_{\substack{\mathbf{k},\mathbf{k}',\mathbf{q}\\ \sigma,\sigma'}} \widetilde{V}_{\sigma\sigma'}(\mathbf{k}-\mathbf{k}')\hat{c}_{\mathbf{k}+\mathbf{q}/2,\sigma}^\dagger \hat{c}_{-\mathbf{k}+\mathbf{q}/2,\sigma'}^\dagger \hat{c}_{-\mathbf{k}'+\mathbf{q}/2,\sigma'} \hat{c}_{\mathbf{k}'+\mathbf{q}/2,\sigma}, \tag{6.1}$$

where $\hat{c}_{\mathbf{k},\sigma}^\dagger$, $\hat{c}_{\mathbf{k},\sigma}$ ($\sigma = \uparrow, \downarrow$) are fermionic creation and annihilation operators and $\widetilde{V}_{\sigma\sigma'}(\mathbf{k}-\mathbf{k}')$ is the interaction matrix element between the two-electron states $|\mathbf{k}+\frac{\mathbf{q}}{2},\sigma; -\mathbf{k}+\frac{\mathbf{q}}{2},\sigma'\rangle$ and $|\mathbf{k}'+\frac{\mathbf{q}}{2},\sigma; -\mathbf{k}'+\frac{\mathbf{q}}{2},\sigma'\rangle$. The assumption that the ground state $|\Phi_{BCS}\rangle$ of the superconductor is a condensate of Cooper pairs implies a nonzero expectation value of the pair amplitude,

$$\Phi_{\mathbf{k}',\mathbf{q}}^{\sigma\sigma'} \equiv \langle \Phi_{BCS} | \hat{c}_{-\mathbf{k}'+\mathbf{q}/2,\sigma'} \hat{c}_{\mathbf{k}'+\mathbf{q}/2,\sigma} | \Phi_{BCS} \rangle \neq 0. \tag{6.2}$$

Similarly to what we did for the Bose gas, we use this fact to write $\hat{c}_{-\mathbf{k}'+\mathbf{q}/2,\sigma'}\hat{c}_{\mathbf{k}'+\mathbf{q}/2,\sigma} = \Phi_{\mathbf{k}',\mathbf{q}}^{\sigma\sigma'} + (\hat{c}_{-\mathbf{k}'+\mathbf{q}/2,\sigma'}\hat{c}_{\mathbf{k}'+\mathbf{q}/2,\sigma} - \Phi_{\mathbf{k}',\mathbf{q}}^{\sigma\sigma'})$ in Eq. (6.1) and retain only terms up to first order in the fluctuations $(\hat{c}_{-\mathbf{k}'+\mathbf{q}/2,\sigma'}\hat{c}_{\mathbf{k}'+\mathbf{q}/2,\sigma} - \Phi_{\mathbf{k}',\mathbf{q}}^{\sigma\sigma'})$. Introducing the *gap parameter*

$$\Delta_{\mathbf{k},\mathbf{q}}^{\sigma\sigma'} \equiv \frac{1}{L^D}\sum_{\mathbf{k}'}\widetilde{V}_{\sigma\sigma'}(\mathbf{k}-\mathbf{k}')\Phi_{\mathbf{k}',\mathbf{q}}^{\sigma\sigma'} \tag{6.3}$$

we thus obtain that the Gaussian approximation for $\hat{\mathscr{H}} = \hat{H} - \mu\hat{N}$ is

$$
\begin{aligned}
\hat{\mathscr{H}}_G = &\sum_{\mathbf{k},\sigma}(\epsilon_{\mathbf{k}}-\mu)\hat{c}_{\mathbf{k},\sigma}^\dagger\hat{c}_{\mathbf{k},\sigma} \\
&+ \frac{1}{2}\sum_{\substack{\mathbf{k},\mathbf{q}\\ \sigma,\sigma'}}\left(-\Delta_{\mathbf{k},\mathbf{q}}^{\sigma\sigma'*}\Phi_{\mathbf{k},\mathbf{q}}^{\sigma\sigma'} + \Delta_{\mathbf{k},\mathbf{q}}^{\sigma\sigma'*}\hat{c}_{-\mathbf{k}+\mathbf{q}/2,\sigma'}\hat{c}_{\mathbf{k}+\mathbf{q}/2,\sigma} + \Delta_{\mathbf{k},\mathbf{q}}^{\sigma\sigma'}\hat{c}_{\mathbf{k}+\mathbf{q}/2,\sigma}^\dagger\hat{c}_{-\mathbf{k}+\mathbf{q}/2,\sigma'}^\dagger\right).
\end{aligned}
\tag{6.4}
$$

To simplify things a bit, we now consider the case of a spin singlet superconductor, where the effective interaction is $\widetilde{V}_{\sigma\sigma'}(\mathbf{k}) = (1-\delta_{\sigma\sigma'})\widetilde{V}(\mathbf{k})$, and we assume that the Cooper pairs have zero center-of-mass momentum, i.e. $\mathbf{q} = \mathbf{0}$. In this case, Eq. (6.4) becomes

$$
\begin{aligned}
\hat{\mathscr{H}}_G &= \sum_{\mathbf{k},\sigma}(\epsilon_{\mathbf{k}}-\mu)\hat{c}_{\mathbf{k},\sigma}^\dagger\hat{c}_{\mathbf{k},\sigma} + \sum_{\mathbf{k}}\left(-\Delta_{\mathbf{k}}^*\Phi_{\mathbf{k}} + \Delta_{\mathbf{k}}^*\hat{c}_{-\mathbf{k},\downarrow}\hat{c}_{\mathbf{k},\uparrow} + \Delta_{\mathbf{k}}\hat{c}_{\mathbf{k},\uparrow}^\dagger\hat{c}_{-\mathbf{k},\downarrow}^\dagger\right) \\
&= -\sum_{\mathbf{k}}\Delta_{\mathbf{k}}^*\Phi_{\mathbf{k}} + \sum_{\mathbf{k}}(\epsilon_{\mathbf{k}}-\mu) + \sum_{\mathbf{k}}\begin{pmatrix}\hat{c}_{\mathbf{k},\uparrow}^\dagger & \hat{c}_{-\mathbf{k},\downarrow}\end{pmatrix}\begin{pmatrix}\epsilon_{\mathbf{k}}-\mu & \Delta_{\mathbf{k}} \\ \Delta_{\mathbf{k}}^* & -\epsilon_{\mathbf{k}}+\mu\end{pmatrix}\begin{pmatrix}\hat{c}_{\mathbf{k},\uparrow} \\ \hat{c}_{-\mathbf{k},\downarrow}^\dagger\end{pmatrix},
\end{aligned}
\tag{6.5}
$$

with

$$\Phi_{\mathbf{k}'} = \langle\Phi_{BCS}|\hat{c}_{-\mathbf{k}',\downarrow}\hat{c}_{\mathbf{k}',\uparrow}|\Phi_{BCS}\rangle, \qquad \Delta_{\mathbf{k}} = \frac{1}{L^D}\sum_{\mathbf{k}'}\widetilde{V}(\mathbf{k}-\mathbf{k}')\Phi_{\mathbf{k}'}. \tag{6.6}$$

The matrix in the last line of Eq. (6.5) is Hermitian and thus can be diagonalized by the unitary Bogoliubov transformation [4]

$$\begin{pmatrix}\hat{d}_{\mathbf{k},\uparrow} \\ \hat{d}_{-\mathbf{k},\downarrow}^\dagger\end{pmatrix} = \begin{pmatrix}u_{\mathbf{k}} & -v_{\mathbf{k}} \\ v_{\mathbf{k}}^* & u_{\mathbf{k}}^*\end{pmatrix}\begin{pmatrix}\hat{c}_{\mathbf{k},\uparrow} \\ \hat{c}_{-\mathbf{k},\downarrow}^\dagger\end{pmatrix}, \qquad \begin{pmatrix}\hat{c}_{\mathbf{k},\uparrow} \\ \hat{c}_{-\mathbf{k},\downarrow}^\dagger\end{pmatrix} = \begin{pmatrix}u_{\mathbf{k}}^* & v_{\mathbf{k}} \\ -v_{\mathbf{k}}^* & u_{\mathbf{k}}\end{pmatrix}\begin{pmatrix}\hat{d}_{\mathbf{k},\uparrow} \\ \hat{d}_{-\mathbf{k},\downarrow}^\dagger\end{pmatrix}, \tag{6.7}$$

where $\hat{d}_{\mathbf{k},\sigma}$, $\hat{d}_{\mathbf{k},\sigma}^\dagger$ are new fermionic operators satisfying canonical anticommutation relations, provided that the complex functions $u_{\mathbf{k}}$, $v_{\mathbf{k}}$ satisfy

$$|u_{\mathbf{k}}|^2 = 1 - |v_{\mathbf{k}}|^2 = \frac{1}{2}\left(1 + \frac{\epsilon_{\mathbf{k}}-\mu}{\mathcal{E}_{\mathbf{k}}}\right), \qquad u_{\mathbf{k}}^*v_{\mathbf{k}} = -\frac{\Delta_{\mathbf{k}}}{2\mathcal{E}_{\mathbf{k}}}, \tag{6.8}$$

where

$$\mathcal{E}_{\mathbf{k}} = \sqrt{(\epsilon_{\mathbf{k}}-\mu)^2 + |\Delta_{\mathbf{k}}|^2}. \tag{6.9}$$

The diagonalized Hamiltonian reads

$$\hat{\mathscr{H}}_G = -\sum_{\mathbf{k}}\Delta_{\mathbf{k}}^*\Phi_{\mathbf{k}} + \sum_{\mathbf{k}}(\epsilon_{\mathbf{k}}-\mu-\mathcal{E}_{\mathbf{k}}) + \sum_{\mathbf{k},\sigma}\mathcal{E}_{\mathbf{k}}\hat{d}_{\mathbf{k},\sigma}^\dagger\hat{d}_{\mathbf{k},\sigma}. \tag{6.10}$$

This shows that the elementary excitations of the system, the Bogoliubov quasiparticles created by $\hat{d}_{\mathbf{k},\sigma}^\dagger$, have a minimum energy equal to $|\Delta_{\mathbf{k}}|$, which is nonzero in the superconducting phase (hence the name gap parameter). Owning to the energy gap separating filled and empty quasiparticle states, these are difficult to excite at low temperatures, implying the rigidity of

the BCS ground state $|\Phi_{BCS}\rangle$. The latter is the vacuum state of the algebra $\{d_{\mathbf{k},\sigma}, \hat{d}_{\mathbf{k},\sigma}^{\dagger}\}$, namely the state that is annihilated by all annihilation operators $\hat{d}_{\mathbf{k},\sigma}$:

$$|\Phi_{BCS}\rangle = \prod_{\mathbf{k}} \hat{d}_{\mathbf{k},\uparrow} \hat{d}_{-\mathbf{k},\downarrow}|0\rangle = \prod_{\mathbf{k}} \left(u_{\mathbf{k}} + v_{\mathbf{k}} \hat{c}_{\mathbf{k},\uparrow}^{\dagger} \hat{c}_{-\mathbf{k},\downarrow}^{\dagger}\right)|0\rangle, \tag{6.11}$$

where $|0\rangle$ is the vacuum state of the the algebra $\{\hat{c}_{\mathbf{k},\sigma}, \hat{c}_{\mathbf{k},\sigma}^{\dagger}\}$.

The grand canonical partition function is then

$$\begin{aligned}
\mathcal{Z}_G &= \mathrm{Tr}\left(e^{-\beta \hat{\mathcal{H}}_G}\right) \\
&= e^{-\beta[-\sum_{\mathbf{k}} \Delta_{\mathbf{k}}^* \Phi_{\mathbf{k}} + \sum_{\mathbf{k}}(\epsilon_{\mathbf{k}} - \mu - \mathcal{E}_{\mathbf{k}})]} \prod_{\substack{\sigma=\uparrow,\downarrow}} \sum_{\{N_{\mathbf{k},\sigma}^{ex}\}} \langle\{N_{\mathbf{k},\sigma}^{ex}\}|e^{-\beta \sum_{\mathbf{k}} \mathcal{E}_{\mathbf{k}} \hat{N}_{\mathbf{k},\sigma}^{ex}}|\{N_{\mathbf{k},\sigma}^{ex}\}\rangle \\
&= e^{-\beta[-\sum_{\mathbf{k}} \Delta_{\mathbf{k}}^* \Phi_{\mathbf{k}} + \sum_{\mathbf{k}}(\epsilon_{\mathbf{k}} - \mu - \mathcal{E}_{\mathbf{k}})]} \prod_{\mathbf{k}} \left(1 + e^{-\beta \mathcal{E}_{\mathbf{k}}}\right)^2,
\end{aligned} \tag{6.12}$$

which gives the grand potential

$$\Omega_G = -\sum_{\mathbf{k}} \Delta_{\mathbf{k}}^* \Phi_{\mathbf{k}} + \sum_{\mathbf{k}} (\epsilon_{\mathbf{k}} - \mu - \mathcal{E}_{\mathbf{k}}) - \frac{2}{\beta} \sum_{\mathbf{k}} \ln\left(1 + e^{-\beta \mathcal{E}_{\mathbf{k}}}\right). \tag{6.13}$$

At this point, the gap parameter $\Delta_{\mathbf{k}}$ and the chemical potential $\mu$ must be determined self-consistently solving a 'gap equation' together with a 'number equation'.

*Gap equation*—Substituting Eq. (6.11) into Eq. (6.6), and making use of Eq. (6.8), we obtain

$$\Phi_{\mathbf{k}'} = \langle 0|(u_{\mathbf{k}'}^* + v_{\mathbf{k}'}^* \hat{c}_{-\mathbf{k}',\downarrow} \hat{c}_{\mathbf{k}',\uparrow}) \hat{c}_{-\mathbf{k}',\downarrow} \hat{c}_{\mathbf{k}',\uparrow} (u_{\mathbf{k}} + v_{\mathbf{k}'} \hat{c}_{\mathbf{k}',\uparrow}^{\dagger} \hat{c}_{-\mathbf{k}',\downarrow}^{\dagger})|0\rangle = u_{\mathbf{k}'}^* v_{\mathbf{k}'} = -\frac{\Delta_{\mathbf{k}'}}{2\mathcal{E}_{\mathbf{k}'}}. \tag{6.14}$$

It follows that the self-consistent equation for the gap parameter at zero temperature is

$$\Delta_{\mathbf{k}} = -\frac{1}{L^D} \sum_{\mathbf{k}'} \widetilde{V}(\mathbf{k} - \mathbf{k}') \frac{\Delta_{\mathbf{k}'}}{2\mathcal{E}_{\mathbf{k}'}}. \tag{6.15}$$

At finite temperature, the anomalous average $\Phi_{\mathbf{k}'}$ is defined on a thermal state rather than on the ground state. Using relations (6.7) and the fact that the only nonzero thermal average for noninteracting quasiparticles is $\langle \hat{d}_{\mathbf{k},\sigma}^{\dagger} \hat{d}_{\mathbf{k},\sigma}\rangle = (e^{\beta \mathcal{E}_{\mathbf{k}}} + 1)^{-1}$ (Fermi-Dirac distribution), we obtain

$$\Phi_{\mathbf{k}'} = \langle \hat{c}_{-\mathbf{k}',\downarrow} \hat{c}_{\mathbf{k}',\uparrow}\rangle = -u_{\mathbf{k}'}^* v_{\mathbf{k}'} \langle \hat{d}_{\mathbf{k}',\uparrow}^{\dagger} \hat{d}_{\mathbf{k}',\uparrow}\rangle + u_{\mathbf{k}'}^* v_{\mathbf{k}'} \langle \hat{d}_{-\mathbf{k}',\downarrow} \hat{d}_{-\mathbf{k}',\downarrow}^{\dagger}\rangle = -\frac{\Delta_{\mathbf{k}'}}{2\mathcal{E}_{\mathbf{k}'}} \tanh\left(\frac{\beta \mathcal{E}_{\mathbf{k}'}}{2}\right), \tag{6.16}$$

and thus the finite-temperature gap equation

$$\Delta_{\mathbf{k}} = -\frac{1}{L^D} \sum_{\mathbf{k}'} \widetilde{V}(\mathbf{k} - \mathbf{k}') \frac{\Delta_{\mathbf{k}'}}{2\mathcal{E}_{\mathbf{k}'}} \tanh\left(\frac{\beta \mathcal{E}_{\mathbf{k}'}}{2}\right). \tag{6.17}$$

This corresponds to the condition

$$\frac{\partial \Omega_G}{\partial \Delta_{\mathbf{k}}^*} = 0. \tag{6.18}$$

*Number equation*—The total number of particles is $N = 2\langle c_{\mathbf{k},\uparrow}^{\dagger} c_{\mathbf{k},\uparrow}\rangle$, where the factor of two accounts for the two spin polarizations. Using again relations (6.7), we obtain

$$N = \sum_{\mathbf{k}} \left[1 - \frac{\epsilon_{\mathbf{k}} - \mu}{\mathcal{E}_{\mathbf{k}}} \tanh\left(\frac{\beta \mathcal{E}_{\mathbf{k}}}{2}\right)\right], \tag{6.19}$$

that is an implicit equation for $\mu$. This corresponds to the thermodynamic relation

$$N = -\frac{\partial \Omega_G}{\partial \mu}. \tag{6.20}$$

**Zero-range interaction**—In the case of an attractive zero-range interaction modeled by a delta function potential $\widetilde{V}(\mathbf{x} - \mathbf{x}') = g\delta^D(\mathbf{x} - \mathbf{x}') = -g_0\delta^D(\mathbf{x} - \mathbf{x}')$, with $g_0 > 0$, we have $\widetilde{V}(\mathbf{k}) = -g_0 = \text{const.}$, hence the gap parameter $\Delta = -\frac{g_0}{L^D}\sum_{\mathbf{k}'}\Phi_{\mathbf{k}'}$ is constant, the gap equation simplifies to

$$\frac{1}{g_0} = \frac{1}{L^D}\sum_{\mathbf{k}}\frac{\tanh(\beta\mathcal{E}_{\mathbf{k}}/2)}{2\mathcal{E}_{\mathbf{k}}}, \tag{6.21}$$

and the grand potential reduces to

$$\Omega_G = \frac{L^D|\Delta|^2}{g_0} + \sum_{\mathbf{k}}(\epsilon_{\mathbf{k}} - \mu - \mathcal{E}_{\mathbf{k}}) - \frac{2}{\beta}\sum_{\mathbf{k}}\ln\left(1 + e^{-\beta\mathcal{E}_{\mathbf{k}}}\right). \tag{6.22}$$

## 6.2 Path integral approach

The same results can be obtained with the path integral. We start directly with the case of a spin singlet superconductor, whose grand canonical partition function is given by [Eq. (2.46)]

$$\mathcal{Z} = \int \mathcal{D}\overline{\Psi}_{\uparrow}\mathcal{D}\Psi_{\uparrow}\mathcal{D}\overline{\Psi}_{\downarrow}\mathcal{D}\Psi_{\downarrow}\,e^{-\frac{1}{\hbar}\int_0^{\beta\hbar}d\tau\int d^D\mathbf{x}\,\mathcal{L}[\overline{\Psi}_{\uparrow},\Psi_{\uparrow},\overline{\Psi}_{\downarrow},\Psi_{\downarrow}]}, \tag{6.23}$$

where [Eq. (6.1)]

$$\mathcal{L} = \sum_{\sigma=\uparrow,\downarrow}\overline{\Psi}_{\sigma}(\mathbf{x},\tau)\left(\hbar\partial_{\tau} - \frac{\hbar^2\nabla^2}{2m} - \mu\right)\Psi_{\sigma}(\mathbf{x},\tau)$$

$$+ \int d^D\mathbf{x}'\,V(\mathbf{x} - \mathbf{x}')\overline{\Psi}_{\uparrow}(\mathbf{x},\tau)\overline{\Psi}_{\downarrow}(\mathbf{x}',\tau)\Psi_{\downarrow}(\mathbf{x}',\tau)\Psi_{\uparrow}(x,\tau). \tag{6.24}$$

is the Lagrangian density of the system, and $\Psi_{\sigma}(\mathbf{x})$ are the Grassmann eigenfunctions associated to the fermionic coherent states $|\Psi\rangle = e^{-\sum_{\sigma=\uparrow,\downarrow}\int d^D\mathbf{x}\,\Psi_{\sigma}(\mathbf{x})\hat{\Psi}_{\sigma}^{\dagger}(\mathbf{x})}|0\rangle$ that we use as the representation basis for the path integral [cf. Eq. (2.42)]. Introducing an auxiliary bilocal bosonic field $\Delta(\mathbf{x}, \mathbf{x}', \tau)$, which will turn out to be the gap parameter, the interaction term can be decoupled via the Hubbard-Stratonovich transformation

$$e^{\frac{1}{\hbar}\iint V(\mathbf{x}-\mathbf{x}')\overline{\Psi}_{\uparrow}(\mathbf{x},\tau)\overline{\Psi}_{\downarrow}(\mathbf{x}',\tau)\Psi_{\downarrow}(\mathbf{x}',\tau)\Psi_{\uparrow}(\mathbf{x},\tau)} =$$

$$\int \mathcal{D}\Delta^*\mathcal{D}\Delta\,e^{-\frac{1}{\hbar}\iint\left[\Delta^*(\mathbf{x},\mathbf{x}',\tau)V^{-1}(\mathbf{x}-\mathbf{x}')\Delta(\mathbf{x},\mathbf{x}',\tau) - \left(\Delta^*(\mathbf{x},\mathbf{x}',\tau)\Psi_{\downarrow}(\mathbf{x}',\tau)\Psi_{\uparrow}(\mathbf{x},\tau) + \text{h.c.}\right)\right]}, \tag{6.25}$$

where $\iint \equiv \int_0^{\beta\hbar}d\tau\int d^D\mathbf{x}\,d^D\mathbf{x}'$. Defining the two-component Nambu spinors

$$\psi(\mathbf{x},\tau) = \begin{pmatrix}\Psi_{\uparrow}(\mathbf{x},\tau)\\\overline{\Psi}_{\downarrow}(\mathbf{x},\tau)\end{pmatrix}, \qquad \overline{\psi}(\mathbf{x},\tau) = \begin{pmatrix}\overline{\Psi}_{\uparrow}(\mathbf{x},\tau) & \Psi_{\downarrow}(\mathbf{x},\tau)\end{pmatrix}, \tag{6.26}$$

we can rewrite the partition function as

$$\mathcal{Z} = \int \mathcal{D}\Delta^*\mathcal{D}\Delta\mathcal{D}\overline{\psi}\mathcal{D}\psi\,e^{-S_{HS}[\Delta^*,\Delta,\overline{\psi},\psi]/\hbar}, \tag{6.27}$$

where

$$S_{HS} = \iint \left[ -\Delta^*(\mathbf{x}, \mathbf{x}', \tau) V^{-1}(\mathbf{x} - \mathbf{x}') \Delta(\mathbf{x}, \mathbf{x}', \tau) + \overline{\psi}(\mathbf{x}, \tau)[-\mathcal{G}^{-1}(\mathbf{x}, \mathbf{x}', \tau)] \psi(\mathbf{x}', \tau) \right], \quad (6.28)$$

$$-\mathcal{G}^{-1}(\mathbf{x}, \mathbf{x}', \tau) = \begin{pmatrix} \delta(\mathbf{x} - \mathbf{x}')(\hbar \partial_\tau - \frac{\hbar^2 \nabla^2}{2m} - \mu) & \Delta(\mathbf{x}, \mathbf{x}', \tau) \\ \Delta^*(\mathbf{x}, \mathbf{x}', \tau) & \delta(\mathbf{x} - \mathbf{x}')(\hbar \partial_\tau + \frac{\hbar^2 \nabla^2}{2m} + \mu) \end{pmatrix}. \quad (6.29)$$

We can now perform the Gaussian path integral over spinor fields to obtain

$$\mathcal{Z} = \int \mathcal{D}\Delta^* \mathcal{D}\Delta \, e^{-S_{eff}[\Delta^*, \Delta]/\hbar}, \quad (6.30)$$

where

$$S_{eff} = \iint \left[ -\Delta^*(\mathbf{x}, \mathbf{x}', \tau) V^{-1}(\mathbf{x} - \mathbf{x}') \Delta(\mathbf{x}, \mathbf{x}', \tau) \right] - \hbar \ln \det \left[ -\mathcal{G}^{-1}(\mathbf{x}, \mathbf{x}', \tau) \right]. \quad (6.31)$$

Up to here everything is exact. We now make the assumptions that the HS field is static and translationally invariant, $\Delta = \Delta(\mathbf{x} - \mathbf{x}')$. The latter corresponds to the assumption we made in Eq. (6.5), that the Cooper pairs have zero center-of-mass momentum. Taking the Fourier transform of Eq. (6.31), and accounting for the time-ordering, then yields

$$S_{eff} = -\beta\hbar \sum_{\mathbf{k}} \Delta_{\mathbf{k}}^* \Phi_{\mathbf{k}} - \hbar \sum_q \ln \det \left[ -\beta \mathcal{G}^{-1}(q) \right], \quad (6.32)$$

where

$$\begin{aligned}
-\mathcal{G}^{-1}(q) &= \begin{pmatrix} -\mathcal{G}_{11}^{-1}(q) e^{i\omega_n 0^+} & -\mathcal{G}_{12}^{-1}(q) \\ -\mathcal{G}_{21}^{-1}(q) & -\mathcal{G}_{22}^{-1}(q) e^{-i\omega_n 0^+} \end{pmatrix} \\
&= \begin{pmatrix} (-i\hbar\omega_n + \epsilon_{\mathbf{k}} - \mu) e^{i\omega_n 0^+} & \Delta_{\mathbf{k}} \\ \Delta_{\mathbf{k}}^* & (-i\hbar\omega_n - \epsilon_{\mathbf{k}} + \mu) e^{-i\omega_n 0^+} \end{pmatrix}, \quad (6.33)
\end{aligned}$$

$\omega_n = (2n+1)\pi/\beta\hbar$, and $q = (\mathbf{k}, n)$. The Gaussian approximation now consists in evaluating the partition function at the saddle point of the HS field:

$$\mathcal{Z}_G = e^{-S_{eff}[\Delta_{\mathbf{k}}^*, \Delta_{\mathbf{k}}]/\hbar}, \qquad \Omega_G = \frac{1}{\beta\hbar} S_{eff}[\Delta_{\mathbf{k}}^*, \Delta_{\mathbf{k}}], \quad (6.34)$$

where $\Delta_{\mathbf{k}}, \Delta_{\mathbf{k}}^*$ satisfy the saddle-point conditions

$$\frac{\partial \Omega_G}{\partial \Delta_{\mathbf{k}}^*} = \frac{\partial \Omega_G}{\partial \Delta_{\mathbf{k}}} = 0. \quad (6.35)$$

These identify the HS field as the gap parameter of the BCS theory, see Eq. (6.18).

Using the fact that $\mathcal{G}_{22}^{-1}(q) = -\mathcal{G}_{11}^{-1}(-q)$, and neglecting the overall minus sign [see the discussion following Eq. (4.32)], we can then follow the same procedure of Eqs. (5.41)-(5.42) to obtain

$$\begin{aligned}
\Omega_G &= -\sum_{\mathbf{k}} \Delta_{\mathbf{k}}^* \Phi_{\mathbf{k}} - \frac{1}{\beta} \sum_q \ln \det \left[ -\beta \mathcal{G}^{-1}(q) \right] \\
&= -\sum_{\mathbf{k}} \Delta_{\mathbf{k}}^* \Phi_{\mathbf{k}} - \frac{1}{\beta} \lim_{\delta \to 0^+} \sum_q \left\{ \ln \left[ -\beta \mathcal{G}_{11}^{-1}(q) \right] - \ln \left[ -\beta \mathcal{G}_{22}^{-1}(q) \right] + \ln \det \left[ -\beta \mathcal{G}^{-1}(q) \right] \right\} e^{i\omega_n \delta}.
\end{aligned}$$

$$(6.36)$$

Using [see Eqs. (4.7), (4.35), and (4.29), respectively]

$$\lim_{\delta \to 0^+} \sum_q \ln \left[ -\beta \mathcal{G}_{11}^{-1}(q) \right] e^{i\omega_n \delta} = \sum_{\mathbf{k}} \ln \left[ 1 + e^{-\beta(\epsilon_{\mathbf{k}} - \mu)} \right], \tag{6.37a}$$

$$\lim_{\delta \to 0^+} \sum_q \ln \left[ -\beta \mathcal{G}_{22}^{-1}(q) \right] e^{i\omega_n \delta} = \sum_{\mathbf{k}} \ln \left[ e^{\beta(\epsilon_{\mathbf{k}} - \mu)} + 1 \right], \tag{6.37b}$$

$$\lim_{\delta \to 0^+} \sum_q \ln \det \left[ -\beta \mathcal{G}^{-1}(q) \right] e^{i\omega_n \delta} = \sum_{\mathbf{k}} \left[ \beta \mathcal{E}_{\mathbf{k}} + 2 \ln \left( 1 + e^{-\beta \mathcal{E}_{\mathbf{k}}} \right) \right], \tag{6.37c}$$

we finally obtain

$$\Omega_G = -\sum_{\mathbf{k}} \Delta_{\mathbf{k}}^* \Phi_{\mathbf{k}} + \sum_{\mathbf{k}} (\epsilon_{\mathbf{k}} - \mu - \mathcal{E}_{\mathbf{k}}) - \frac{2}{\beta} \sum_{\mathbf{k}} \ln \left( 1 + e^{-\beta \mathcal{E}_{\mathbf{k}}} \right), \tag{6.38}$$

which coincides with the Hamiltonian result, Eq. (6.13).

**Alternative time-ordering**—Let us consider also the alternative time-ordering, in which both components of the matrix (6.33) appear with the same convergence factor. As described in Section 4.2, the action becomes

$$S_{eff} = -\beta \hbar \sum_{\mathbf{k}} \Delta_{\mathbf{k}}^* \Phi_{\mathbf{k}} + \beta \hbar \sum_{\mathbf{k}} (\epsilon_{\mathbf{k}} - \mu)$$
$$- \hbar \sum_q \ln \det \left[ \beta \begin{pmatrix} -\mathcal{G}_{11}^{-1}(q) e^{i\omega_n 0^+} & -\mathcal{G}_{12}^{-1}(q) \\ -\mathcal{G}_{21}^{-1}(q) & -\mathcal{G}_{22}^{-1}(q) e^{i\omega_n 0^+} \end{pmatrix} \right]. \tag{6.39}$$

The corresponding grand potential is

$$\Omega_G = -\sum_{\mathbf{k}} \Delta_{\mathbf{k}}^* \Phi_{\mathbf{k}} + \sum_{\mathbf{k}} (\epsilon_{\mathbf{k}} - \mu) - \frac{1}{\beta} \sum_q \ln \det \left[ -\beta \mathcal{G}^{-1}(q) \right] e^{i\omega_n 0^+}, \tag{6.40}$$

and using result (6.37c) for the last summation, we immediately obtain Eq. (6.38).

We emphasize that here (differently from the case of the weakly-interacting Bose gas, where the chemical potential is fixed by the condensate density) obtaining exactly this grand potential is crucial; if we had computed $\det[-\beta \mathcal{G}^{-1}(q)]$ naively, as in Eq. (5.47), the second term in Eq. (6.38) would have been $-\sum_{\mathbf{k}} \mathcal{E}_{\mathbf{k}}$, and consequently the relation $N = -\partial \Omega_G / \partial \mu$ would have produced an incorrect number equation.

# 7  Conclusion

We have presented a unified account of the coherent-state path-integral approach to the partition function of quantum many-particle systems, emphasizing the technical subtleties that, if overlooked, can lead to incorrect or inconsistent results. Starting from the construction of the discretized path integral, we showed how to take the continuum limit in imaginary time or in Matsubara-frequency space without losing contact with the canonical Hamiltonian formalism. Through a sequence of paradigmatic examples, from the bosonic and fermionic harmonic oscillators to the weakly interacting Bose gas and the BCS superconductor, we demonstrated that, when treated with due care, the path-integral formalism reproduces exactly the thermodynamic quantities obtained by canonical operator methods. Our emphasis has been deliberately technical rather than interpretive, focusing on how to compute equilibrium quantities rather than on the physical consequences of the results, which are extensively reviewed in the existing literature. We hope these notes serve as a practical reference and a didactic complement to popular textbooks, and that they help readers avoid common pitfalls while applying coherent-state path integrals to new problems.

**Acknowledgments**    The authors thank Lorenzo Frigato, Giacomo Gradenigo, Adam Rançon, and Jacques Tempere for useful comments.

**Funding information**    L.S. and C.V. are supported by the Project "Frontiere Quantistiche" (Dipartimenti di Eccellenza) of the Italian Ministry of University and Research (MUR) and by "Iniziativa Specifica Quantum" of INFN. L.S. is partially supported by funds of the European Union-Next Generation EU: European Quantum Flagship Project "PASQuanS2", National Center for HPC, Big Data and Quantum Computing [Spoke 10: Quantum Computing], and he also acknowledges the PRIN Project "Quantum Atomic Mixtures: Droplets, Topological Structures, and Vortices" of MUR.

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
