# Peer review of "Coherent-state path integrals in quantum thermodynamics"

_SciPost Physics Lecture Notes_

## Round 1 · Referee Report · Anonymous (Referee 1) · 2025-11-26

Report

In this paper the authors review some subtleties with the coherent-state path integral expression for the partition function of non-relativistic quantum matter. As far as I can see the paper does not contain any new material that has not already be covered in the textbooks by, for instance, Kleinert, Negele & Orland, and Stoof et al. to name just a few. Moreover, instead of properly referencing this literature the authors seem to imply that these subtleties are treated here for the first time, which is just not fair in my opinion and does not do justice to the existing textbooks that have treated all the topics of this review already in quite some detail and have clearly explained the problems involved.

Recommendation

Reject

---

## Round 1 · Referee Report · Adam Rancon (Referee 2) · 2025-12-20

Strengths

1- pedagogical discussion of subtleties in coherent-state path integral

Weaknesses

1- Somewhat unfocused: some subsections are textbook material that are not related to the subtleties of path integrals.

Report

The manuscript discusses pedagogically the subtleties that arise when dealing with coherent state path integrals. These include the issue of non-linear changes of variables, which can lead to wrong results when not performed properly; the fact that Hubbard-Stratonovich fields are white noise and should be handled properly in functional determinants; the problem of convergence factors in Matusbara sum. The first two issues have been the subject of some debates in the recent literature, and could deserve being the subject of a review or of some pedagogical notes. This is much more debatable concerning the last issue, since it is well-known and discussed in many textbooks.

Furthermore, the manuscript is somewhat unfocused, since additional textbook material unrelated to coherent state path integrals are discussed at length (sec. 5.1.3, 5.2, 6.3, App A and B). These sections should be removed.

Sometimes, the authors don't seem to quite know the level of pedagogy they aim for. For instance, they want to start at a very basic level, but do not give the canonical (anti)-commutation relations of the operators, which seems strange. See also my remarks below.

I think the manuscript could be published if it were more focused and expanded on the subtleties discussed in the recent literature, and after my comments below are addressed, and the requested changes are implemented.

1- I don't quite agree with the discussion of Q_k(eps) around Eq. 2.4. In Eq. 2.4, all terms of order greater than 1 in eps are neglected (eps standing for $\tilde \epsilon$), not just the ones in Q_k(eps). See the (correct) discussion of the functional determinant, which is computed with an error of order 1/M^2~eps^2.

2- page 12, I do not understand what the discussion of mean-field vs saddle point brings as it is written. Furthermore, I have an issue with the calculation done in Eq. 3.41: there, N is NOT a HS field behaving as a white noise, so there is no reason why there should be a correction term g/2 in the exponent exp(beta(mu+g/2-Ng). Furthermore, Eq. 3.39 is ambiguous: what is S_MF in that equation? It is not even clear what calculation is done. Do they use the HS field phi, do the integration over the a,a^*, and then do the saddle-point approximation? If so, then extra g/2 in the exponent is correct. Otherwise, it is not clear. My suggestion is that, after the calculations done on that page are corrected/clarified, a discussion of the difference between MF and SP would be interesting, and, it seems to me, new. This is a question left open in the conclusion of Ref. 14 that would be interesting to address here.

3- In the fermionic case (p.13), there does not seem to be an Ito correction term. Why is it so? It is unfortunate that the authors do not discuss the discretized version to explain why it appears to be so different from the bosonic case. This is again a problem that has not been addressed in the literature and deserves a pedagogical discussion here.

4- When dealing with convergence factors for Matsubara sums, the authors do not do much better than what most textbooks usually do: they drop the convergence factors that appear at the discretized level, to add them back when necessary, with a sign only chosen to reproduce the correct result. This is sloppy and goes against the advertised pedagogical purpose of these notes. The sentence "‘reg’ reminds us that the divergent summation should be properly regularized. Here ‘properly’ means, of course, that the result should give back Eq. (4.7)" is in this respect particularly damaging. If the authors want to keep this section, it should be written in a way to avoid these kinds of pitfalls.

  • Similarly, as it is written, Sec. 5 and 6 are superfluous, as the handling of convergence factors is not done better than in textbooks (it is just a rehash of, e.g., chapter 7 of Dupuis'). For instance, the sentence "The treatment we have presented, which is based on the careful introduction of convergence factors motivated by the implicit time ordering of the path integral" is misleading, since the convergence factors have disappeared in Eq. 5.37, and are added by hand in 5.44 to obtain the correct result. If they were kept throughout the calculation, there would be at least a pedagogical interest. The same goes for the BCS calculation.

Requested changes

1 - remove sec. 5.1.3, 5.2, 6.3, App A and B

3- Sec. 2: the canonical (anti)-commutation relations should be given. The fact that fermionic operators anticommute with Grassmann variables should be discussed. The latter should be defined.

3- Eq. 3.23: H(N) should be defined right away. It is implicit, and can very well be confusing (since it looks very much like \hat H, and could imply that it is \hat H with \hat N replaced by N, which it is not!)

4- Bosonic and fermionic Matsubara frequencies should be defined around eq. 4.2, not in Sec. 4.2

5- after eq. 4.16: The number N can be set by taking the limit beta->infty, which unbigously set it to 0.

6- below Eq. 5.2, L is not defined, and the dilute limit should read a_s^D N/L^D<<1.

7- p. 31: I do not understand the equation i pi lim_{delta->0^+} Sum_n e^{-i omega_n delta}=0, nor its connection to Eq. 4.27

Recommendation

Ask for major revision

---

## Round 1 · Referee Report · Jacques Tempere (Referee 3) · 2025-12-23

Strengths

In this paper Salasnich and Vianello are our pilots, guiding us safely past the treacherous sandbanks and dangerous rapids of the path-integral formulation of quantum field theory. The goal is to set up a description of interacting Bose and Fermi gases, including superfluidity. The authors provide a lot of details that are usually swept under the rug, and cause confusion in students (and lecturers).

For example, the explanation of the convergence factors (and their origin) for the Matsubara summations in section 4 is very enlightening. I admit that
the additional beta*(hbar omega/2) term that appears erroneously in some textbooks has been puzzling me, and here I find it unambiguously and clearly explained.

Report

This pedagogical and detailed explanation deserves publication in scipost physics lecture notes. I am convinced it will be very useful to many people, and that it is of current interest.

Requested changes

The authors can consider a few small comments. Firstly, it was necessary for me to remind myself that Grassman variables not only anticommute among each other, but also anticommute with their operators. That appears when having the annihilation operator act on the right hand side of 2.14, in order to obtain 2.15a. It might be useful to remind other readers of this as well.

A step that may deserve a bit more attention is the one going from (3.20), the second-quantized version of the Hamiltonian, to (3.22), the Euclidean action. In the last term of (3.22). In relation to the problem identified in the section below, one could wonder why there is (a* a)^2 to start with, rather than (a*a)(a*a-1). Usually, the problem is formulated the other way around, with a quantization scheme (for example Weyl quantization) specifying how to assign operators to phase space variables. Since there are different choices possible, I think it would be clarifying if the authors gave more explanation at this point.

Another interesting detail that could be commented on is the \omega=0 frequency term for bosons. When using the \omega_(-n) = - \omega_n symmetry as in (4.22) care must be taken for that special term that is the only one not doubled by this symmetry. It would be nice if the authors could add a discussion of that detail as errors there also lead in an additional term not unlike the one they address already.

Recommendation

Publish (easily meets expectations and criteria for this Journal; among top 50%)

---

## Editorial Decision

awaiting_resubmission